



# Comparison of ECHAM5/MESSy Atmospheric Chemistry (EMAC) Simulations of the Arctic winter 2009/2010 and 2010/2011 with Envisat/MIPAS and Aura/MLS Observations

Farahnaz Khosrawi[1], Oliver Kirner[2], Gabriele Stiller[1], Michael Höpfner[1], Michelle L. Santee[3], Sylvia Kellmann[1], and Peter Braesicke[1]

[1]Institute of Meteorology and Climate Research, Karlsruhe Institute of Technology, Karlsruhe, Germany
[2]Steinbuch Centre for Computing, Karlsruhe Institute of Technology, Karlsruhe, Germany
[3]Jet Propulsion Laboratory, California Institute of Technology, California, USA

*Correspondence to:* Farahnaz Khosrawi (farahnaz.khosrawi@kit.edu)

**Abstract.** We present model simulations with the atmospheric chemistry-climate model ECHAM5/MESSy Atmospheric Chemistry (EMAC) nudged toward European Center for Medium-Range Weather Forecasts (ECMWF) reanalyses for the Arctic winter 2009/2010 and 2010/2011. This study is the first to perform an extensive assessment of the performance of the EMAC model for Arctic winters; previous studies have only made limited evaluations of EMAC simulations for the Arctic. We

have chosen the two extreme Arctic winters 2009/2010 and 2010/2011 to evaluate the formation of polar stratospheric clouds (PSCs) and the representation of the chemistry and dynamics of the polar winter stratosphere in EMAC. The EMAC simulations are compared to observations by the Michelson Interferometer for Passive Atmospheric Soundings (Envisat/MIPAS) and the observations from the Aura Microwave Limb Sounder (Aura/MLS). The Arctic winter 2010/2011 was one of the coldest winters on record, leading to the strongest depletion of ozone measured in the Arctic. The Arctic winter 2009/2010 was, from

the climatological perspective, one of the warmest winters on record. However, it was distinguished by an exceptionally cold stratosphere (colder than the climatological mean). Cold temperatures prevailed from mid December 2009 to mid January 2010, leading to prolonged PSC formation and existence. Significant denitrification, the removal of $HNO_3$ from the stratosphere by sedimentation of $HNO_3$ containing polar stratospheric cloud particles, occurred. In our comparison, we focus on polar stratospheric cloud formation and denitrification. The comparisons between EMAC simulations and satellite observations show that

model and measurements compare well for these two Arctic winters (differences for $HNO_3$ generally within $\pm 20\%$) and thus that EMAC nudged toward ECMWF reanalyses is capable of giving a realistic representation of the evolution of PSCs and the associated sequestration of gas-phase $HNO_3$ in the polar winter stratosphere. However, simulated PSC volume densities are several orders of magnitude smaller than the ones derived from Envisat/MIPAS observations. Further, PSCs in EMAC are not simulated as high up as they are observed. This underestimation of PSC volume density and vertical extension of the PSCs

results in an underestimation of the vertical redistribution of $HNO_3$ due to denitrification/re-nitrification.



# 1 Introduction

The severity of ozone destruction during polar winter in the lower stratosphere is dependent on the prevailing meteorology. During cold Arctic winters temperatures are sufficiently low to allow for the formation of polar stratospheric clouds (PSCs). PSCs play a key role in stratospheric ozone destruction (Solomon et al., 1986; Crutzen and Arnold, 1986). In the polar spring,

heterogeneous reactions take place on and within the PSC particles which convert halogens from relatively inert reservoir species into forms which can destroy ozone efficiently (e.g. Peter, 1997; Solomon, 1999; Lowe and MacKenzie, 2008).

PSCs consist of liquid or solid particles. According to their composition and physical state PSCs have been classified into three different types: (1) supercooled ternary solutions (STS), (2) Nitric Acid Trihydrate (NAT) and (3) ice. Liquid PSC cloud particles (STS) form by the condensation of water vapour ($H_2O$) and nitric acid ($HNO_3$) on the liquid stratospheric

background sulfate aerosol particles at temperatures 2–3 K below the NAT existence temperature $T_{NAT}$ ($\sim 195$ K at 50 hPa). For the formation of solid cloud particles (NAT and ice) much lower temperatures (slightly above or below the ice frost point $T_{ice}$ ($\sim 188$ K at 50 hPa) are required (e.g. Carslaw et al., 1994; Koop et al., 1995). PSCs form at altitudes between 15-30 km. Denitrification, the permanent removal of $HNO_3$ by sedimenting polar stratospheric cloud particles, limits the deactivation process of the ozone destroying substances in springtime and thus leads to a prolongation of the ozone destroying catalytic

cycles.

The Arctic winters 2009/2010 and 2010/2011 were both exceptional. The Arctic winter 2010/2011 was one of the most persistently cold winters on record, leading to the strongest depletion of ozone measured in the Arctic (Manney et al., 2011). The Arctic winter 2010/2011 has been well analysed, especially with respect to ozone loss (e.g. Manney et al., 2011; Sinnhuber et al., 2011; Arnone et al., 2012; Kuttipurath et al., 2012; Hommel et al., 2014). The dynamical perspective, thus the excep-

tional dynamical conditions of this winter, were discussed in detail by Hurwitz et al. (2011); Isaksen et al. (2012); Strahan et al. (2013); Shaw and Perlwitz (2014). Although the Arctic winter 2009/2010 was one of the warmest winters on record (Dörnbrack et al., 2012), it was distinguished by an exceptionally cold stratosphere (colder than the climatological mean) from mid December 2009 to mid January 2010, leading to prolonged PSC formation and significant denitrification (e.g. Khosrawi et al., 2011). The Arctic winter 2009/2010 has been well analysed both by measurements and model simulations. Detailed studies on

e. g. denitrification during this winter were performed by e. g. Khosrawi et al. (2011) and on dehydration by e. g. Khaykin et al. (2013). An overview on the dynamical situation during this winter is given by Dörnbrack et al. (2012). Measurements were performed within the project RECONCILE (Reconciliation of essential parameters for an enhanced predictability of Arctic stratospheric ozone loss and its climate interactions), an intensive field campaign with the M55-Geophysica research aircraft. An overview of the measurements and the results derived within this project is given in von Hobe et al. (2013).

Due to their importance in processes leading to ozone destruction in the polar winter stratosphere, an accurate representation of PSCs and binary sulphuric acid/water (background) aerosols is essential for the correct simulation of chlorine activation and polar ozone depletion in chemistry-climate models (CCMs). Here, we compare simulations of the Arctic winter stratosphere made with the submodel MSBM (formerly PSC (Kirner et al., 2011, 2015)) of the chemistry-climate model ECHAM5/MESSy Atmospheric Chemistry (EMAC, (Jöckel et al., 2006, 2010)) with observations from the Michelson Interferometer for Passive



Atmospheric Soundings (Envisat/MIPAS) (Fischer and Oelhaf, 1996; Fischer et al., 2008) and the Aura Microwave Limb Sounder (Aura/MLS) (Waters et al., 2006). The submodel MSBM (Multi-phase stratospheric box model) has been tested and optimised for the simulations of PSCs in the Antarctic (Kirner et al., 2011, 2015). EMAC simulation results of stratospheric nitrogen compounds and ozone were evaluated with Envisat/MIPAS observations by Brühl et al. (2007) for the Antarctic winter
5  2002.

The previous studies had their focus on the Antarctic, especially on ozone and chlorine activation. So far, no EMAC evaluation study has been performed focusing on the Arctic stratosphere. In this study, we test the representativeness of the EMAC simulations of PSCs and related quantities in the Arctic on the basis of simulations for the Arctic winters 2009/2010 and 2010/2011. In our study the focus is on PSC volume density, temperature and $HNO_3$. Further (qualitative) comparisons with
additional trace gases ($O_3$, ClO and $H_2O$) were performed in another study for the Arctic winter 2015/2016 (Khosrawi et al., 2017) where EMAC simulations were compared with observations from Aura/MLS and the Gimballed Limb Observer for Radiance Imaging of the Atmosphere (GLORIA) performed on board of the High Altitude and LOng-range Research Aircraft (HALO).

## 2   Model Simulations and Observations

### 2.1   The chemistry-climate model EMAC

The ECHAM/MESSy Atmospheric Chemistry (EMAC) model is a numerical chemistry and climate simulation system that includes submodels describing tropospheric and middle atmosphere processes and their interaction with oceans, land and human influences (Jöckel et al., 2010). It uses the second version of the Modular Earth Submodel System (MESSy2) to link multi-institutional computer codes. The core atmospheric model is the 5th generation European Centre Hamburg general
circulation model (ECHAM5, Roeckner et al. (2006)). For the present study we applied EMAC (ECHAM5 version 5.3.02, MESSy version 2.50) in T63L90MA resolution, i.e., with a spherical truncation of 63 (corresponding to a quadratic Gaussian grid of approximately $1.875° \times 1.875°$ degrees in latitude and longitude) with 90 vertical hybrid pressure levels from the surface up to $0.01\,hPa$ (approx. $80\,km$). A Newtonian relaxation technique of the prognostic variables temperature, vorticity, divergence and the (logarithm of the) surface pressure above the boundary layer and below $1\,hPa$ towards European Center of
Medium-Range Weather Forecast (ECMWF) ERA-Interim reanalysis data (Dee et al., 2011) was applied, in order to nudge the model dynamics towards the observed meteorology.

Here we analyse a T63L90 EMAC simulation that was chemically initialised based on a previous simulation performed for the time period 1994 to 2010. The simulation was started on 1 January 2009 and continued until December 2011. The simulation includes a comprehensive chemistry setup for the stratosphere and troposphere. Reaction rate coefficients for gas
phase reactions and absorption cross sections for photolysis are taken from Atkinson et al. (2007) and Sander et al. (2011b). The applied model setup comprised the submodels: ONEMIS for "online" emissions of tracers and aerosols, OFFEMIS for "offline" emissions of tracers and aerosols, TNUDGE for tracer nudging (Kerkweg et al., 2006a), DDEP for dry deposition of trace gases and aerosols, SEDI for the sedimentation of aerosol particles (Kerkweg et al., 2006b), MECCA for the gas-phase



chemistry (Sander et al., 2011a), JVAL for the calculation of photolysis rates (Sander et al., 2014), SCAV for the scavenging and liquid phase chemistry in cloud and precipitation (Tost et al., 2006a), CONVECT for the parameterisation of convection (Tost et al., 2006b), LNOX for the source of NOx produced by lightning (Tost et al., 2007b), MSBM (Multi-phase Stratospheric Box Model) for the processes related to polar stratospheric clouds (Kirner et al., 2011), PTRAC for additional prognostic tracers

(Jöckel et al., 2008), CVTRANS for convective tracer transport (Tost et al., 2006b), TROPOP for diagnosing the tropopause and boundary layer height (Jöckel et al., 2006), SORBIT for sampling model data along sun-synchronous satellite orbits (Jöckel et al., 2010), H2O for stratospheric water vapor (Jöckel et al., 2006), RAD for the radiation calculation (Jöckel et al., 2016), and CLOUD for calculating the cloud cover as well as cloud microphysics including precipitation (Tost et al., 2007a).

The submodel MSBM simulates the number densities, mean radii and surface areas of sulphuric acid aerosols and the

different polar stratospheric cloud particles (STS, NAT and ice). The formation of STS particles is calculated through the up-take of $HNO_3$ and $H_2O$ on the liquid binary sulphuric acid/water particles (Carslaw et al., 1995). Ice particles are assumed to form homogeneously at temperatures below the ice frost point temperature. For the calculation of the formation of solid particles a thermodynamic approach based on the equilibrium between the gas phase and the solid phase is assumed (e.g. Chipperfield, 1999). The vapour pressure over ice and NAT are calculated according to the parameterisations given in Marti

and Mauersberger (1993) and Hanson and Mauersberger (1988), respectively. For the simulation of NAT particles two parame-terisations are included in the submodel MSBM, one based on the heterogeneous formation of NAT on ice ("thermodynamical NAT parameterisation") and one based on the homogeneous formation of NAT ("kinetic growth NAT parameterisation"). The "thermodynamical" NAT parameterisation assumes an instantaneous thermodynamical equilibrium while the "kinetic" param-eterisation is based on the growth and sedimentation algorithm given by Carslaw et al. (2002) and van den Broek et al. (2004).

We use the kinetic growth parameterisation in our EMAC simulation. NAT formation takes place as soon as a supercooling of 3 K below the NAT existence temperature ($T_{NAT}$) is reached. A comprehensive description of the submodel MSBM can be found in Kirner et al. (2011).

## 2.2 Envisat/MIPAS

MIPAS is a middle infrared Fourier transform spectrometer. MIPAS was launched in March 2002 onboard Envisat and was op-

erational until the sudden loss of contact with Envisat on 8 April 2012. MIPAS measured the atmospheric emission spectrum in the limb sounding geometry. MIPAS operated in its nominal observation mode from June 2002 to March 2004. Measurements during this time period were performed in its full spectral resolution measurement mode with a designated spectral resolution of $0.035\,cm^{-1}$. Measurements were performed covering the altitude range from the mesosphere to the troposphere with a high vertical resolution (about 3 km in the stratosphere). After a failure of the interferometer slide at the end of March 2004, MIPAS

resumed measurements in January 2005 with a reduced spectral resolution of $0.0625\,cm^{-1}$, but with improved spatial resolu-tion. Data products of Envisat/MIPAS include 30 trace species, e.g. $H_2O$, $O_3$, $HNO_3$, $CH_4$, $N_2O$, $NO_2$ as well as temperature (Fischer and Oelhaf, 1996; Fischer et al., 2008). Here, the Envisat/MIPAS $HNO_3$ and temperature data version V5H_HNO3_20 and V5H_T_20 and V5R_HNO3_224/225 and V5R_T_220/221 (NOMINAL mode) derived with the IMK/IAA retrieval pro-cessor covering the periods July 2002–March 2003 and January 2005–April 2012, respectively, have been used (updated





version of the retrieval as described in Milz et al. (2009) and von Clarmann et al. (2009)). Comparison of the MIPAS $HNO_3$ data product with satellite measurements from Odin/SMR (Odin/Sub-Millimetre Radiometer), ADEOS/ILAS-II (Advanced Earth Observing Satellite/Improved Limb Atmospheric Spectrometer), SCISAT/ACE-FTS (SCISAT/Atmospheric Chemistry Experiment-Fourier Transform Spectrometer) as well as the Envisat/MIPAS ESA (European Space Agency) product showed

good agreement and differences were generally within $\pm 0.5$ ppbv (Wang et al., 2007; Wolff et al., 2008). Differences between MIPAS $HNO_3$ and the balloon-borne version of MIPAS (MIPAS-B) and the infrared spectrometer MkIV were even smaller, with less than 0.5 ppbv. However, differences can increase up to 1–2 ppbv between 22 and 26 km at high latitudes due to the large horizontal inhomogeneity of $HNO_3$ near the vortex boundary (Wang et al., 2007). Further comparisons of MIPAS data with other observations as well as application of the MIPAS $HNO_3$ data to Antarctic studies can be found in Stiller et al. (2005)

and Mengistu Tsidu et al. (2005). In addition to the $HNO_3$ and temperature data, the MIPAS PSC Data V5R_PSCVD_120_220 is used. This new MIPAS retrieval product of PSC volume density profiles has been derived recently (Höpfner et al., 2018, in preparation). The retrieval procedure is similar to that described in Höpfner et al. (2006) but optimized such that the effect of different PSC types and sizes (e. g. Höpfner, 2004) on the estimated volume density is minimized. For each MIPAS limb-scan this data set provides two profiles of PSC volume densities on a standard altitude grid of 1 km. These profiles indicate the range

of possible values of volume density which are compatible with the MIPAS radiances under different assumptions for particle size. The typical vertical resolution of the retrieval is about 3–4 km.

## 2.3   Aura/MLS

The Microwave Limb Sounder (MLS) on the Earth Observing System Aura satellite was launched in July 2004. The Aura/MLS instrument is an advanced successor to the MLS instrument on the Upper Atmosphere Research satellite (UARS). MLS is

a limb sounding instrument that measures the thermal emission at millimetre and submillimetre wavelengths using seven radiometers to cover five broad spectral regions (Waters et al., 2006). Measurements are performed from the surface to 90 km with a daily global latitude coverage from 82° S to 82° N. Here, we use Aura/MLS version v4.2 $HNO_3$ and temperature data. The data screening criteria given by Livesey et al. (2017) have been applied to the data. A detailed assessment of the quality and reliability of the Aura/MLS v2.2 $HNO_3$ measurements can be found in Santee et al. (2007). The $HNO_3$ in v4.2

has been significantly improved compared to v2.2. In particular, the low bias in the stratosphere has been largely eliminated. Measurements of v4.2 $HNO_3$ are performed with a horizontal resolution of 400–500 km and a vertical resolution of 3–4 km over most of the vertical range. In the lower stratosphere, the precision and systematic uncertainty for $HNO_3$ are estimated to be 0.6 ppbv and 1.0-1.5 ppbv (2-$\sigma$ estimates), respectively (Livesey et al., 2017). The MLS v4.2x temperatures are similar to both the v3.3 and to the v2.2 temperatures described in the validation study by Schwartz et al. (2008). MLS v4.2 temperatures have

a $\sim -1$ K bias with respect to correlative measurements in the troposphere and stratosphere. Further, in the MLS temperatures persistent, vertically oscillating biases with respect to analysis and correlative measurements have been found in the troposphere and stratosphere (see Livesey et al. (2017) for more details).



# 3 Arctic winter 2009/2010 and 2010/2011

In the following a short description of the characteristics of the 2009/2010 and 2010/2011 Arctic winter considered in this study will be given. The Arctic winter 2010/2011 was one of the most persistently cold winters on record. The prevailing cold temperatures lead to the strongest depletion of ozone measured in the Arctic. The Arctic winter 2009/2010 on the other

hand was rather warm in the climatological sense, but was distinguished by an exceptionally cold stratosphere from mid December 2009 to mid January 2010 that lead to prolonged PSC formation and significant denitrification (Dörnbrack et al., 2012; Khosrawi et al., 2011).

## 3.1 Arctic winter 2009/2010

The polar vortex formed in December and a Canadian warming in mid-December caused a splitting of the vortex into two parts.

The colder part of the vortex was located over the Canadian Arctic and survived, resulting in a vortex recovery. The vortex then further cooled down through mid-January so that temperatures sufficiently low to allow solid PSCs to form were reached. The polar stratosphere was unusually cold from mid-December 2009 to end of January 2010. A comparison of ECMWF temperatures of the Arctic winters of the past half century showed that the 2009/2010 Arctic winter was one of the few winters with synoptic-scale temperatures below $T_{ice}$ (Pitts et al., 2011). Additionally, during mid-January orographic waves

were frequently excited by the flow over Greenland. A major warming in the second half of January (around 24th January) caused a displacement of the vortex to the European Arctic and also initiated the breakup of the vortex (Pitts et al., 2011; Dörnbrack et al., 2012).

   In Pitts et al. (2011) a detailed description as well as examples of the PSCs observed by CALIPSO during the Arctic winter 2009/2010 is given. The measurements of PSCs by CALIPSO during that winter can be divided into four phases with distinctly

different PSC optical characteristics: (1) 15–30 December 2009: The first phase was dominated by patchy, tenuous clouds consisting of liquid/NAT mixtures. (2) 31 December 2009 to 14 January 2010: The second phase was characterized by the occurrence of mountain wave ice clouds along the east coast of Greenland, enhanced numbers of Mix-2 and Mix-2 particles, as well as fully developed liquid STS clouds. (3) 15 to 21 January 2010: The third distinct phase occurred when temperatures synoptically cooled below $T_{ice}$, resulting in synoptic-scale ice PSCs. (4) 22 to 28 January 2010: The fourth and last phase was

dominated by liquid STS clouds (Pitts et al., 2011).

## 3.2 Arctic winter 2010/2011

The Arctic winter 2010/2011 was one of the coldest within the last two decades (Manney et al., 2011; Sinnhuber et al., 2011) and was characterised by an anomalously strong vortex with an atypically long cold period that was persistent from mid-December to mid-March (Manney et al., 2011). The polar vortex formed at the end of November 2010 and remained stable

until the end of April. Due to minor warmings, the long cold period, lasting over four months, was interrupted by short warmer periods in the beginning of January, February and March. In February and March, temperatures were colder than in previous



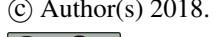

years of the last decade. The final warming during the 2010/2011 Arctic winter occurred later than usual, in mid-April (Arnone et al., 2012; Kuttipurath et al., 2012).

Based on CALIPSO measurements the PSC season during the 2010/2011 winter can be divided into four PSC phases according to the four cold phases that occurred over the four month period from December 2010 to March 2011. The time

periods of these four phases and the PSC types that occurred during each phase are as follows (Khosrawi et al., 2012): (1) 23 December 2010 to 8 January 2011: STS, Mix 1/2 and ice clouds. (2) 20–28 January 2011: mainly Mix-1 and Mix-2 with some STS and ice. (3) 5–27 February: STS, Mix-1 and Mix-2 as well as ice clouds. (4) 5–19 March: STS clouds (Note: no CALIPSO data is available from 8 to 13 March).

## 4  Model - Measurement comparisons

For our analysis the EMAC simulation with T63L90 resolution nudged toward ECMWF reanalyses is used. The simulation was started on 1 January 2009 and continued until 31 December 2011. PSCs are calculated using the submodel MSBM (see Sect. 2.1). For the formation of NAT we used the kinetic NAT parameterisation, thus NAT particles are formed as soon as the temperature drops below $T_{\mathrm{NAT}} - 3\,\mathrm{K}$. The relative percentage differences between EMAC and the satellite measurements have been calculated by using the following equation: $\mathrm{D} = [\mu(\mathrm{EMAC}) - \mu(\mathrm{Satellite})]/\{[\mu(\mathrm{EMAC}) + \mu(\mathrm{Satellite})]/2\} * 100$ with

D in percent (%), where $\mu$ denotes the mixing ratio (e.g. $HNO_3$) for EMAC and satellite measurements. The comparison is performed for two Arctic winters, 2009/2010 and 2010/2011. These two winters were quite different in their characteristics (Sect. 3) and are thus perfectly suited for testing the model performance concerning chemistry and dynamics of the Arctic winter stratosphere. For the comparison of the $HNO_3$ distribution in daily maps the 34 hPa level is used, while for the comparison of $HNO_3$ time series the 50 hPa level is used. These levels were chosen because these were the respective levels

where $HNO_3$ gas phase depletion was most pronounced.

### 4.1  Simulation of the Arctic winter 2009/2010 and comparison to Envisat/MIPAS and Aura/MLS observations

Figure 1 shows the gas-phase distribution of $HNO_3$ as simulated with EMAC for selected dates between 21 December 2009 and 29 January 2010 at 34 hPa ($\sim$21 km). On 21 December 2009 the $HNO_3$ distribution seems to be almost unchanged compared to the pre-winter distribution (not shown), thus $HNO_3$ values are still high in the polar region. PSC formation has

just started, but not much $HNO_3$ has been taken up by the particles so far and thus removed from the gas phase. From early January onwards the $HNO_3$ values start to decrease and reach very low values north of Scandinavia in mid January (13 and 15 January), indicating a removal from the gas-phase at 34 hPa ($\sim$21 km). Due to the major warming and accompanying dissolution of PSCs, $HNO_3$ values start to increase again toward the end of January.

In Figure 2 the $HNO_3$ distribution from Envisat/MIPAS is shown for the same dates as in Fig. 1. The EMAC simulation of

the Arctic winter 2009/2010 compares generally well to the observations from Envisat/MIPAS. The same holds when EMAC is compared to Aura/MLS (not shown). As in the EMAC simulation the observed $HNO_3$ gas phase distribution shows no signs of $HNO_3$ removal on 21 December. This date falls into the first phase of PSC formation. During this time period (15-30



December) only patchy, tenuous clouds were observed by CALIPSO (Sect. 3.1) which did not significantly affect the $HNO_3$ distribution at 34 hPa. Significant occurrence of PSCs was observed by CALIPSO from January onwards. Accordingly, gas-phase removal of $HNO_3$ in PSCs is found in the Envisat/MIPAS observations from early January onwards and peaks towards mid January. The simulation by EMAC shows the gas-phase depletion in the same areas as observed by Envisat/MIPAS.

However, there are some minor differences between the EMAC simulations and the measurements from Envisat/MIPAS. On 15 January the $HNO_3$ values simulated by EMAC are not as low as those observed by Envisat/MIPAS. Similar differences between EMAC and Envisat/MIPAS are found on 29 January 2010. Also here the EMAC simulation shows somewhat higher $HNO_3$ mixing ratios in the depleted area than was observed by Envisat/MIPAS.

In the following time-pressure cross sections of temperature and $HNO_3$ are considered to investigate how the EMAC sim-
ulations compare to the satellite observations during the course of the winter. Here, measurements from both satellites, Envisat/MIPAS and Aura/MLS, are considered. Figure 3 shows the temporal evolution of temperature and $HNO_3$ from EMAC, Envisat/MIPAS and Aura/MLS for the Arctic winter 2009/2010 (December 2009 to April 2010) at high latitudes (averaged over 70-90°N) as a function of pressure. The patterns of the temporal evolution of $HNO_3$ derived from Envisat/MIPAS and Aura/MLS are generally similar, however, Envisat/MIPAS provides somewhat higher $HNO_3$ abundances than Aura/MLS. Note
that the different sampling properties of the two instruments (with MIPAS not sensitive to gas phase constituents in the presence of PSCs) should be taken into account.

The temperatures from EMAC nudged toward ECMWF are in good agreement with the temperatures measured by Envisat/MIPAS and Aura/MLS. However, the warming in mid January is simulated slightly more strongly in EMAC. Warm temperatures propagate further down into the stratosphere than observed by Envisat/MIPAS and Aura/MLS. Temperatures be-
low 195 K occur in the region between 10 and 50 hPa (~20-28 km) from mid December to mid January. In the simulated and observed $HNO_3$ distribution gas-phase $HNO_3$ is removed from the beginning of January onwards and remains low until end of January.

Although EMAC and the satellite observations from Envisat/MIPAS and Aura/MLS show removal of gas-phase $HNO_3$ in accordance with the time and pressure levels where temperatures below 195 K are found, there are nevertheless obvious differ-
ences between model simulations and satellite observations. One prominent difference between EMAC $HNO_3$ and the satellite observations is that EMAC shows a 2 ppbv lower maximum value in the $HNO_3$ background distribution (early December). Otherwise the early winter $HNO_3$ distribution of EMAC compares well to Envisat/MIPAS and Aura/MLS concerning structure and temporal development. The same holds for the late winter $HNO_3$ distribution. However, this comparison shows that EMAC $HNO_3$ seems generally 1-2 ppbv too low throughout the winter. These too low $HNO_3$ mixing ratios in the EMAC simulation
are also visible in the time series discussed below (Fig. 6 and Fig. 12).

Another prominent difference between EMAC and the observations by Envisat/MIPAS and Aura/MLS is found with the onset of $HNO_3$ gas-phase removal by PSCs and throughout the PSC season (thus throughout January). Here, the satellite observations show $HNO_3$ gas-phase removal which extends from the top of the high background values downwards. A downward propagating tongue of high $HNO_3$ is found below the depleted area, indicating vertical redistribution of $HNO_3$ due to
denitrification/re-nitrification. That the $HNO_3$ gas-phase depletion in the Arctic winter 2009/2010 was permanent and thus led



to a denitrification of the stratosphere in mid-January was shown by e.g. Khosrawi et al. (2011). In the EMAC simulation the denitrified area is shifted towards lower pressure levels and the simulated re-nitrification is not as strong as observed. This difference between EMAC and the satellite observations of Envisat/MIPAS and Aura/MLS in $HNO_3$ throughout the PSC season arises because the simulated PSCs in EMAC differ from those observed by Envisat/MIPAS. This will be discussed in more
detail in the following comparison of PSC volume densities.

Figure 4 and Figure 5 show the temporal evolution of the volume density of PSC particles ($V_{PSC}$ in $\mu m^3 cm^{-3}$) and of NAT and ice particles ($V_{NAT}$ and $V_{ice}$ in $\mu m^3 cm^{-3}$) as a function of pressure as simulated with EMAC.[1] In Figure 4 additionally to the EMAC simulations the total PSC volume density $V_{PSC}$ derived from Envisat/MIPAS is shown. The MIPAS PSC data set provides two profiles of PSC volume densities that indicate the range of possible values, thus an upper and lower limit
(see Sect.2.2). In the Envisat/MIPAS observations the largest PSC volume density is found in January and reaches its maximum in mid January. This is in agreement with PSC observations by CALIPSO during that winter (Sect. 3.1 and references therein). PSCs in EMAC start to form in mid December (mainly NAT). In the beginning of January temperatures drop sufficiently low that also ice formation is briefly simulated.

The EMAC volume density compares well in terms of timing and structure to the volume density from Envisat/MIPAS. How-
ever, the EMAC PSC volume density is several orders of magnitude smaller than the PSC volume density from Envisat/MIPAS (factor of 3 compared to the lower limit and 6-7 compared to the upper limit). This indicates that the amount of liquid particles is underestimated by EMAC. NAT formation in EMAC is already initiated at $T_{NAT}$-3K, which is a rather high threshold temperature for NAT formation. This is usually the temperature where STS formation is initiated since for the formation of NAT particles usually lower temperatures than $T_{NAT}$-3K are required (Peter and Grooß, 2012; Lambert et al., 2016). However, this
approach is commonly applied for simulating PSCs in atmospheric models (e.g. Wohltmann et al., 2013). As was discussed by Wohltmann et al. (2013) this approach technically results in NAT clouds being formed before STS and thus NAT formation occurs at the expense of STS since the NAT clouds consume all the available $HNO_3$. This therefore results in an overestimation of NAT since in reality STS and NAT clouds are often observed at the same time (e.g. Pitts et al., 2011; Peter and Grooß, 2012).

Another difference between EMAC and MIPAS is that EMAC does not simulate PSCs as high as they were observed with
Envisat/MIPAS. This difference is also found when EMAC is compared to CALIPSO, where PSCs are found up to $28\,km$ ($\sim 10\,hPa$, see Fig 9 in Pitts et al. (2011)), thus $HNO_3$ in our EMAC simulation is not removed from the gas-phase as high up as seen in the Envisat/MIPAS and Aura/MLS observations (Fig. 3). Strong re-nitrification is visible in the Envisat/MIPAS and Aura/MLS observations in mid to late January, but is not as strongly simulated with EMAC. This underestimation of re-nitrification in EMAC could be caused by the formation of too many small NAT particles that do not grow to the particle sizes
that are needed to cause significant denitrification and thus re-nitrification below. To better understand the differences between simulated and observed PSCs, further studies are necessary which are however beyond the scope of this study, but will be the subject of future studies.

---

[1]The liquid particles have the largest volume density and thus the liquid volume density $V_{liquid}$ is almost identical to $V_{PSC}$. Therefore, only the total (liquid+solid) PSC volume $V_{PSC}$ density is shown here.



In the following the time series of $HNO_3$ for the Arctic winter 2009/2010 (at 50 hPa, averaged over 70-90°N) derived from the EMAC simulation is compared with the one derived from the Aura/MLS observations (Fig. 6). Both time series show the high background $HNO_3$ mixing ratios that are found at the beginning of the winter. Towards mid January $HNO_3$ mixing ratios decrease to 6 and 8 ppbv, respectively and then start to increase again after the final warming at the end of January. Due

to denitrification during this winter (e.g. Khosrawi et al., 2011) the $HNO_3$ mixing ratios remain lower than they were at the beginning of the winter before the start of the PSC season.

The absolute (not shown) and relative differences between the EMAC and Aura/MLS time series were calculated (Fig. 6). The comparison of the time series confirms the differences found when comparing the cross sections. EMAC $HNO_3$ mixing ratios are generally 1-2 ppbv or up to 20% lower than the ones observed by Aura/MLS during the PSC season (beginning of

December to mid January). The largest relative differences (up to 45%) are found in mid January when the PSCs have their peak occurrence.

## 4.2   Simulation of the Arctic winter 2010/2011 and comparison to Envisat/MIPAS and Aura/MLS observations

As for the Arctic winter 2009/2010 we start with a comparison of daily maps of $HNO_3$. Figure 7 shows the gas-phase distribution of $HNO_3$ as simulated with EMAC for selected dates between 21 December 2010 and 28 March 2011 at 34 hPa

(∼21 km). On 21 December 2010 the $HNO_3$ distribution is still unperturbed by PSC formation. Onset of $HNO_3$ removal is visible from 8 January onwards. On 26 January $HNO_3$ mixing ratios reach a minimum (not shown). On 25 February and 28 March $HNO_3$ mixing ratios have slightly recovered, but remain lower than the pre-winter mixing ratios.

The $HNO_3$ distribution as simulated with EMAC compares generally well with the $HNO_3$ distribution measured during that winter by Envisat/MIPAS (Fig. 8). However, as was found for the Arctic winter 2009/2010, EMAC $HNO_3$ background mixing

ratios tend to be ∼2 ppbv lower than observed and removal of $HNO_3$ seems to be not as strong in the simulation as in the observations. Also during the Arctic winter 2010/2011 $HNO_3$ was permanently removed from the stratosphere, thus leading to a strong denitrification of the Arctic stratosphere as has been shown in previous studies using observations from Aura/MLS, Odin/SMR and Envisat/MIPAS (Manney et al., 2011; Khosrawi et al., 2012; Arnone et al., 2012). Thus, this indicates that in the EMAC simulation of the Arctic winter 2010/2011 also denitrification/re-nitrification is underestimated as will be discussed

further below.

Differences in the daily maps that may be caused by transport processes in the model are found on 8 January and 3 February 2011. The Envisat/MIPAS (Fig. 8) and Aura/MLS (not shown) measurements show an elongated area with low $HNO_3$ stretching over the pole from north to south. In EMAC this area is simulated to be rather circular and thus has a smaller spatial extent than observed. On 3 February 2011 the EMAC simulation shows a dipole structure with two $HNO_3$ minima, one over Russia

and one over Greenland which is not found in the satellite observations. In the Envisat/MIPAS and Aura/MLS observations only the minimum over Russia is observed. Around these two dates the stratosphere was dynamically quite active (Sect. 3.2). Minor warmings disturbed the stratosphere and the differences in $HNO_3$ seen here are most probably related to these. The minor warmings are simulated by EMAC with the correct timing but with a slightly different strength from that observed by Envisat/MIPAS and Aura/MLS as can be seen from the temperature distribution shown in Fig. 9. The cause of these differences





may be related to model dynamics. In fact, an already known feature in EMAC is that the downward transport is underestimated (Brühl et al., 2007; Khosrawi et al., 2009).

Fig. 9 shows the temporal evolution of temperature and $HNO_3$ from EMAC, Envisat/MIPAS and Aura/MLS for the Arctic winter 2010/2011 (December 2010 to April 2011) at high latitudes (70-90°N) as function of pressure. As for the Arctic

winter 2009/2010 the temperatures from EMAC nudged toward ECMWF compare well with the temperatures measured by Envisat/MIPAS and Aura/MLS. However, the warmings in mid January and mid March are slightly more strongly simulated in EMAC than observed by Envisat/MIPAS. Another difference between the simulation and observations is that the area of temperatures lower than T<205 K extends further down in the EMAC simulation than in Envisat/MIPAS observation (especially December to February) while the areas with temperatures T<195 hPa are shifted to higher pressure levels (lower altitudes).

This difference in temperature probably also contributes to the differences between observed and measured PSC (see below) and the underestimation of $HNO_3$ removal we found in the EMAC simulation when comparing the daily maps.

As for the Arctic winter 2009/2010 a prominent difference between EMAC and the Envisat/MIPAS $HNO_3$ for the Arctic winter 2010/2011 is that EMAC exhibits a ~2 ppbv lower maximum mixing ratio in the $HNO_3$ distribution (early December). However, contrary to the 2009/2010 winter here the EMAC $HNO_3$ mixing ratios are slightly higher compared to Aura/MLS.

Otherwise the structure of the early winter $HNO_3$ distribution of EMAC is generally in agreement with Envisat/MIPAS and Aura/MLS. Larger differences (as already noted for the Arctic winter 2009/2010) are found throughout the PSC season. These differences are most probably caused by the differences between the simulated PSCs and those observed by Envisat/MIPAS (Figure 10). Figure 10 and Figure 11 show the temporal evolution of the volume density of PSC particles ($V_{PSC}$ in $\mu m^3 cm^{-3}$) and of NAT and ice particles ($V_{NAT}$ and $V_{ice}$ in $\mu m^3 cm^{-3}$) as a function of pressure as simulated with EMAC. In Figure 10

additionally to the EMAC simulations the total PSC volume density $V_{PSC}$ derived from Envisat/MIPAS is shown.

In Envisat/MIPAS observations and the EMAC simulations the PSC season starts in mid December 2010 and lasts until mid March 2011. In accordance with the four cold phases (Sect. 3.2), four phases of PSCs are observed by Envisat/MIPAS. These four PSC phases are also simulated with the correct timing by EMAC. In early winter the majority of particles simulated with EMAC are NAT since NAT formation is initiated at $T_{NAT}$-3K. Liquid STS particles are found in the EMAC simulation from

mid January to mid March and ice particles at the end of January. The four PSC seasons as simulated with EMAC (concerning time of occurrence of the PSCs and respective PSC types) are in agreement with CALIPSO (see Sect. 3.2 and Pitts and Poole (2014)) and Envisat/MIPAS (concerning time of occurrence of PSCs). However, again PSCs are not simulated as high up as measured by CALIPSO and Envisat/MIPAS. In the beginning of e.g. January PSCs are measured by CALIPSO up to 30 km. In EMAC PSCs are only found up to 15 hPa (approx. 26 km). The Envisat/MIPAS PSC volume also extends over a larger vertical

range than the EMAC PSC volume, indicating (as in the Arctic winter 2010/2011) an underestimation of the PSC occurrence in the EMAC simulation. Another major difference between the EMAC and Envisat/MIPAS volume density is that the EMAC PSC volume is several orders of magnitude smaller than Envisat/MIPAS (factor of 3–4 compared to the lower limit and 5 compared to the upper limit).

The above discussed differences in the PSC volume are also reflected in the temporal evolution of the $HNO_3$ distribution

during the course of the winter (Fig. 9). A prominent difference between the $HNO_3$ distribution simulated by EMAC and



the HNO$_3$ distributions measured by Envisat/MIPAS and Aura/MLS is that EMAC exhibits a slightly smaller background distribution and that the maximum in HNO$_3$ is located at slightly higher pressure levels (lower altitudes). As PSCs form, gas-phase HNO$_3$ is removed, but this removal occurs earlier and more strongly in EMAC, yet restricted to 100 to 50 hPa, while the strong removal of HNO$_3$ seen in the Envisat/MIPAS and Aura/MLS measurements from mid-January onwards extends over

the area from 100 to 20 hPa. Further, the re-nitrification at $\approx$70 hPa (January) which is clearly seen in Envisat/MIPAS and Aura/MLS is not as clearly visible in the EMAC simulation, thus indicating an underestimation of re-nitrification in EMAC.

Similar differences in HNO$_3$ as discussed above are found when comparing the time series derived from EMAC with the one derived from Aura/MLS (at 50 hPa, average over 70-90°N, Fig. 12). In terms of the temporal evolution the time series are in good agreement, but EMAC HNO$_3$ is 1-2 ppbv lower during the time period when PSCs are present as was already

found for the Arctic winter 2009/2010. At the beginning of the winter the HNO$_3$ mixing ratios are in the range of about 10-11 ppbv and then decrease during the course of the winter to 6 ppbv due to sequestration in PSCs and denitrification. The relative differences are generally less than 20% except for the time period when PSC occurrence has its maximum. Here, the differences increase briefly up to 40%.

## 5  Conclusions

We simulated the Arctic winters 2009/2010 and 2010/2011 with the chemistry climate model EMAC and compared the results to satellite observations from Envisat/MIPAS and Aura/MLS. We have chosen these two winters since both winters were quite extreme, but nevertheless different in their chemical and dynamical characteristics. Thus, these two winters are perfectly suited for testing the EMAC performance concerning chemistry and dynamics of the Arctic winter stratosphere. Previous similar but more limited studies focused on the Antarctic; our study is the first to perform such an extensive assessment on the EMAC

performance in the Arctic. The EMAC simulation used in this study was performed with a T63L90 resolution. A Newtonian relaxation technique of the prognostic variables temperature, vorticity, divergence and surface pressure towards ERA-Interim reanalyses was applied below 1 hPa, in order to nudge the model dynamics towards meteorological analyses.

The model simulations for the Arctic winters 2009/2010 and 2010/2011 compare well with the measurements from Envisat/MIPAS and Aura/MLS, showing that EMAC is capable of giving a realistic representation of the Arctic winter strato-

sphere in terms of PSC formation and the HNO$_3$ and temperature distribution. Especially for the (nudged) temperature a very good agreement between satellite measurements and model simulations was found throughout the winter season. However, the warmings were stronger in the simulations by EMAC than observed. The cause of these differences may be related to model dynamics. In fact, a well known feature in EMAC is that the downward transport is underestimated in the lower parts of the polar vortices (Brühl et al., 2007; Khosrawi et al., 2009), despite the model vorticity and divergence fields being nudged

towards ECMWF analyses. Further, as was discussed by Brühl et al. (2007) a too tight subtropical barrier causing too low N$_2$O and too high NO$_y$ in the middle stratosphere is simulated by EMAC. This indicates a need for further improvements of the model dynamics, particularly the forcing by gravity waves (Brühl et al., 2007).





Larger differences than for temperature were found concerning PSC formation/occurrence and the respective gas-phase distribution of $HNO_3$. Here, the comparison between the PSC volume density as simulated with EMAC and the one derived from Envisat/MIPAS observations showed that the simulated PSC volume densities are several orders of magnitude smaller than the observed ones ($\sim$3-7 times smaller in 2009/2010 and $\sim$3-5 times smaller in 2010/2011). Since the PSC volume density

is mainly determined by the liquid PSC volume density, this indicates an underestimation of liquid STS particles in the model. As commonly applied for simulating PSCs in atmospheric models (e.g. Wohltmann et al., 2013), NAT formation in EMAC is initiated at $T_{NAT}$-3K. This results in NAT clouds being formed before STS and thus NAT formation occurs at the expense of STS since the NAT clouds consume all the available $HNO_3$. This therefore results in an overestimation of NAT since in reality STS and NAT clouds are often observed at the same time (e.g. Pitts et al., 2011; Peter and Grooß, 2012). Another difference

between EMAC and Envisat/MIPAS PSC volume densities is that the EMAC PSCs have a smaller vertical extent than those observed by Envisat/MIPAS.

The differences we found in PSC volume densities are reflected in the $HNO_3$ distribution. In 2009/2010 denitrification/re-nitrification is underestimated by EMAC and the denitrified areas are shifted to lower pressure levels. Similar results are derived for the Arctic winter 2010/2011. For this winter PSCs are also not simulated as high up as measured and re-nitrification is un-

derestimated while denitrification occurs earlier and more strongly in EMAC than observed by Envisat/MIPAS and Aura/MLS. This underestimation of re-nitrification in EMAC may be caused by the formation of too many small NAT particles that do not grow to the particle sizes that are needed to cause significant denitrification and thus re-nitrification below. EMAC $HNO_3$ mixing ratios seem generally to be underestimated by 1-2 ppbv. Considering $HNO_3$ time series at 50 hPa we found that differences in $HNO_3$ between EMAC and the observations from Aura/MLS were generally less than 10-20%. However, during the peak

of the PSC season larger differences between EMAC simulations (briefly reaching up to 40%) were found. The comparisons presented here shows that further sensitivity runs are necessary to understand and improve the simulation of Arctic PSCs in EMAC. Sensitivity simulations or adjustments based on observations e.g. concerning the partitioning between STS and NAT and/or the limit for the NAT number density as done by Brakebusch et al. (2013) and Wegner et al. (2013) would help to find the best model set-up for simulating Arctic PSCs. Further, the here derived results and upcoming sensitivity simulations will serve

as benchmark for the development of the PSC paramterisation in other atmospheric models as e. g. ICON-ART (ICOsahedral Nonhydrostatic Model - Aerosols and Reactive Trace gases).

*Acknowledgements.* We would like to thank the European Centre for Medium-Range Weather Forecasts (ECMWF) for providing their meteorological analyses. MLS data were obtained from the NASA Goddard Earth Sciences and Information Center. Work at the Jet Propulsion Laboratory, California Institute of Technology, was done under contract with the National Aeronautics and Space Administration. We are

grateful to T. von Clarmann for helpful discussions. EMAC simulations were performed on the Institute Cluster II at the Steinbuch Center for Computing at Karlsruhe Institute of Technology. We acknowledge support by Deutsche Forschungsgemeinschaft and Open Access Publishing Fund of Karlsruhe Institute of Technology.





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




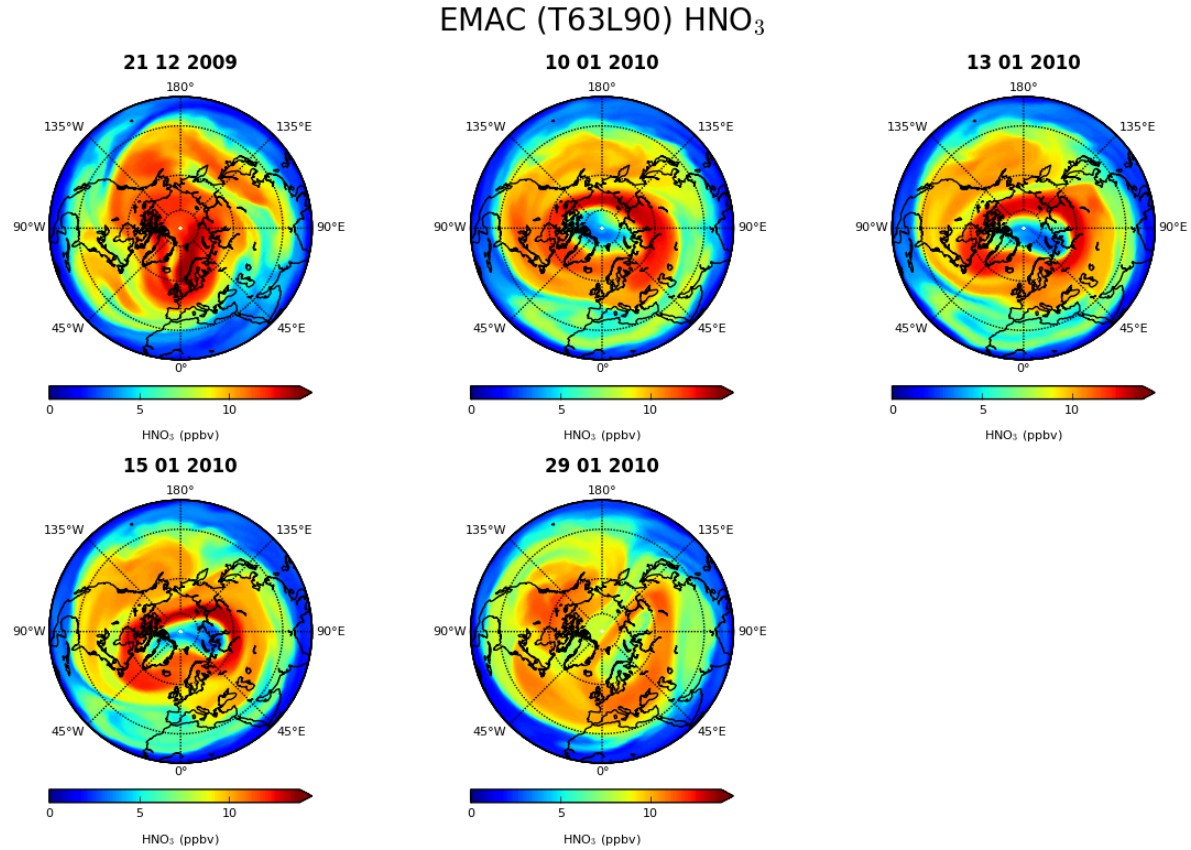

**Figure 1.** Distribution of $HNO_3$ as simulated with EMAC for 21 December 2009, 10, 13, 15 and 29 January 2010 at $34\,\mathrm{hPa}$ ($\approx21\,\mathrm{km}$).




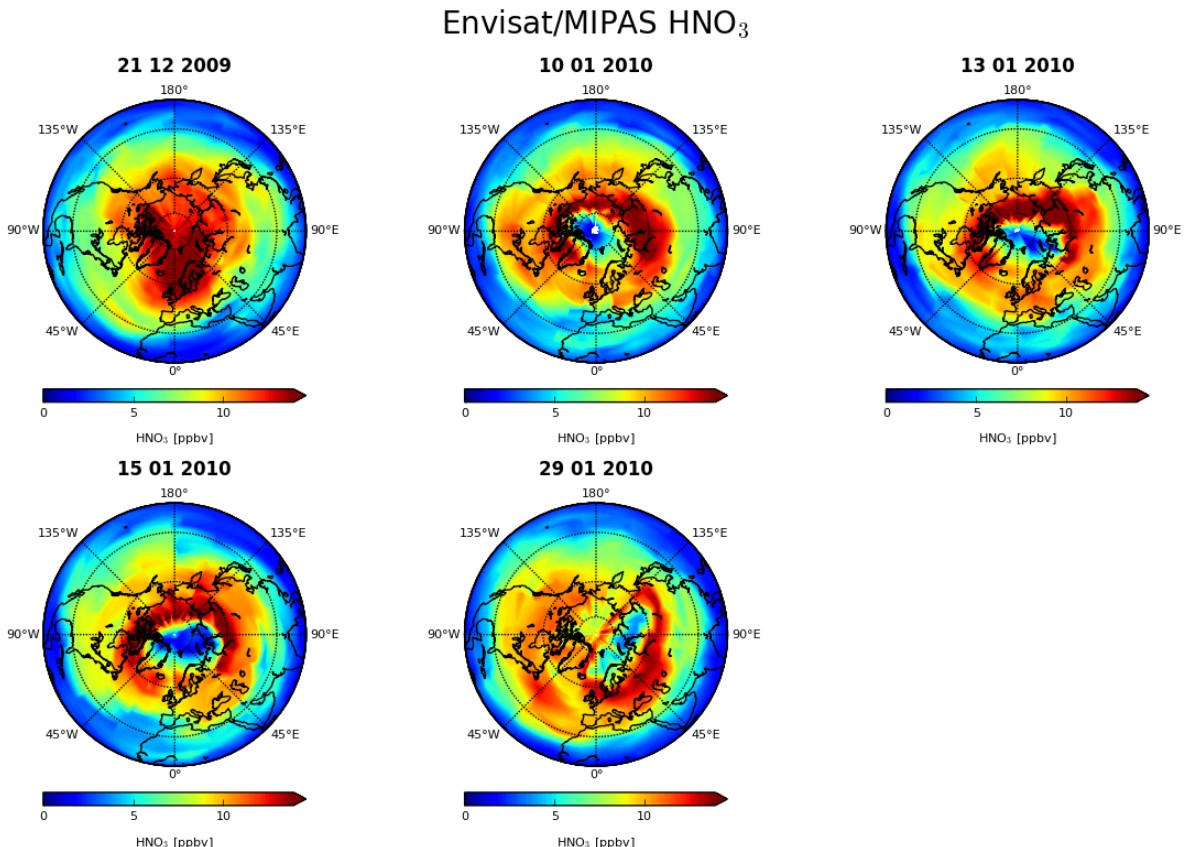

**Figure 2.** Distribution of $HNO_3$ as observed by Envisat/MIPAS on 21 December 2009, 10, 13, 15 and 29 January 2010 at at 21 km ($\sim$34 hPa).



**Figure 3.** Temporal evolution of temperature and HNO$_3$ at northern high latitudes (70-90°N) as function of pressure during the Arctic winter 2009/2010 (December 2009 to April 2010). Top: EMAC, middle: Envisat/MIPAS, bottom: Aura/MLS.



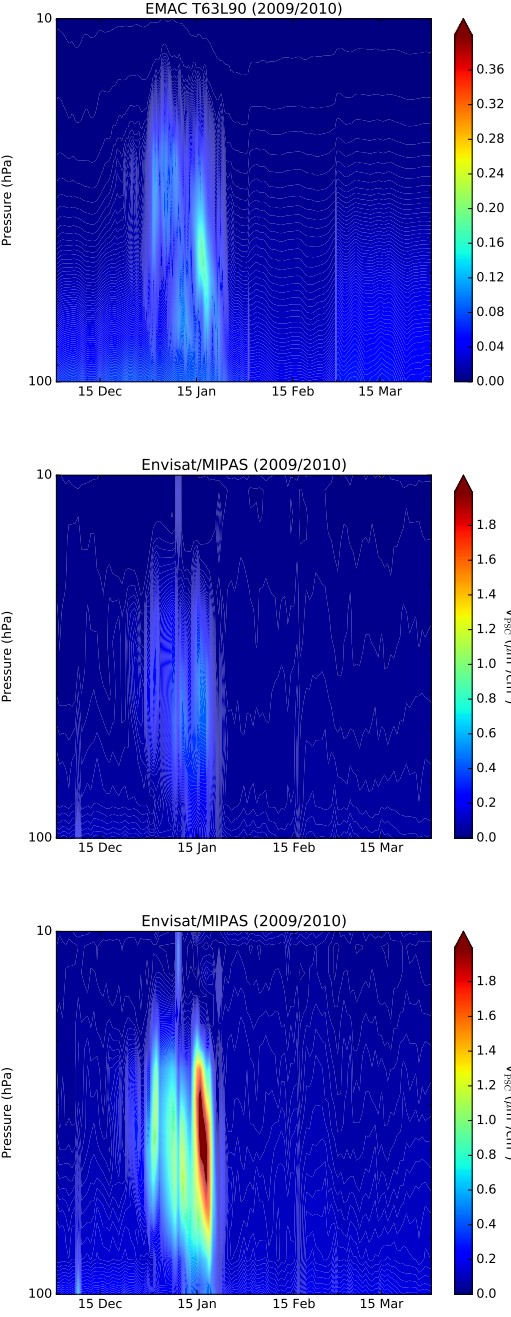

**Figure 4.** Volume density of all (liquid and solid) PSC particles ($V_{PSC}$) in $\mu m^3 cm^{-3}$ as simulated with EMAC (top) and observed with Envisat/MIPAS lower limit (middle) and Envisat/MIPAS upper limit (bottom) for the Arctic winter 2009/2010. Note the different colour bar scales between the Envisat/MIPAS and EMAC panels.





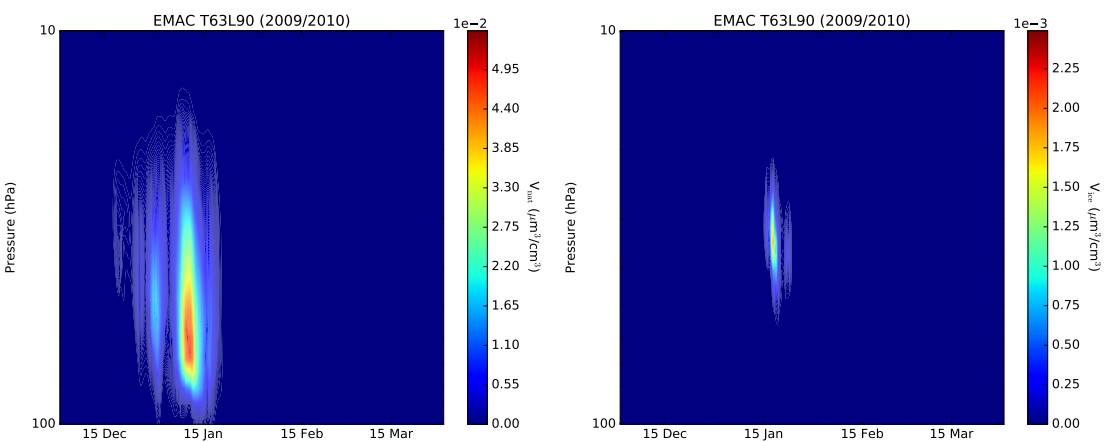

**Figure 5.** Volume density of solid particles ($V_{NAT}$ and $V_{ice}$) in $\mu m^3 cm^{-3}$ as simulated with EMAC for the Arctic winter 2009/2010.



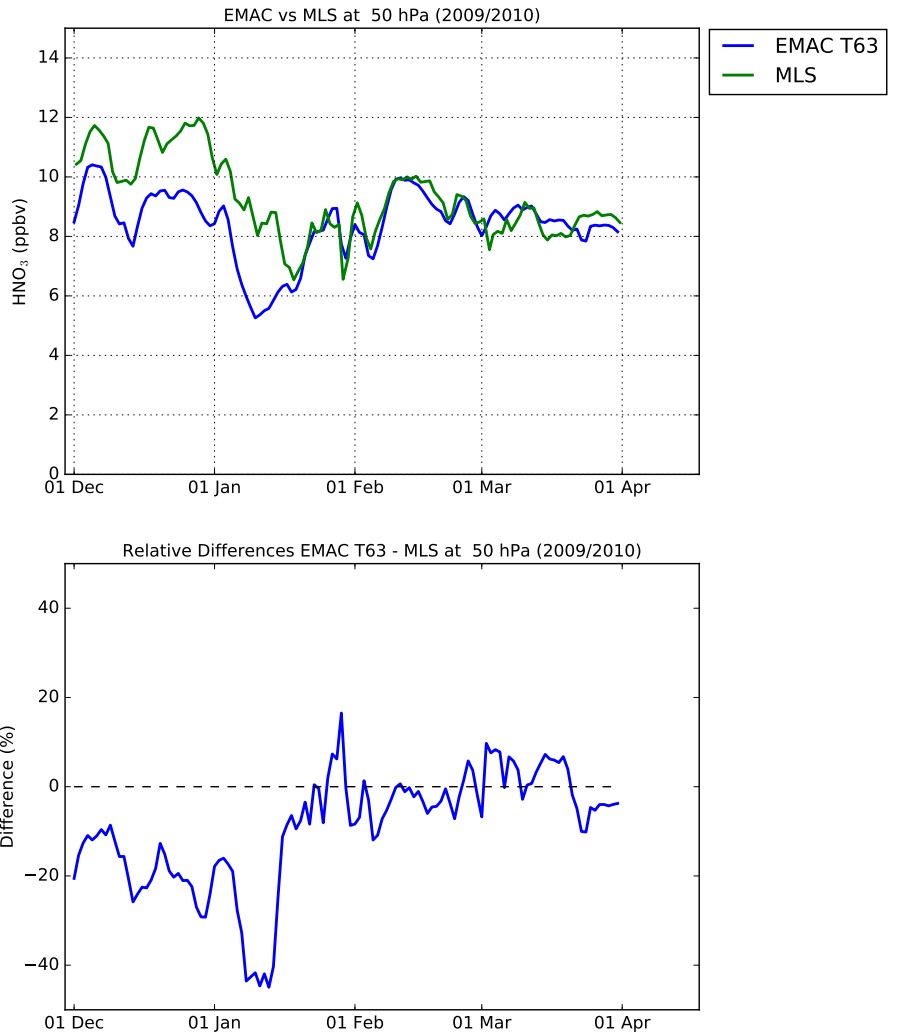

**Figure 6.** Top: Time series of $HNO_3$ (at 50 hPa) from EMAC (blue) and MLS (green) for the Arctic winter 2009/2010 (1 December 2009 to 30 April 2010, 70-90°N). Bottom: Relative differences of the EMAC - MLS time series.



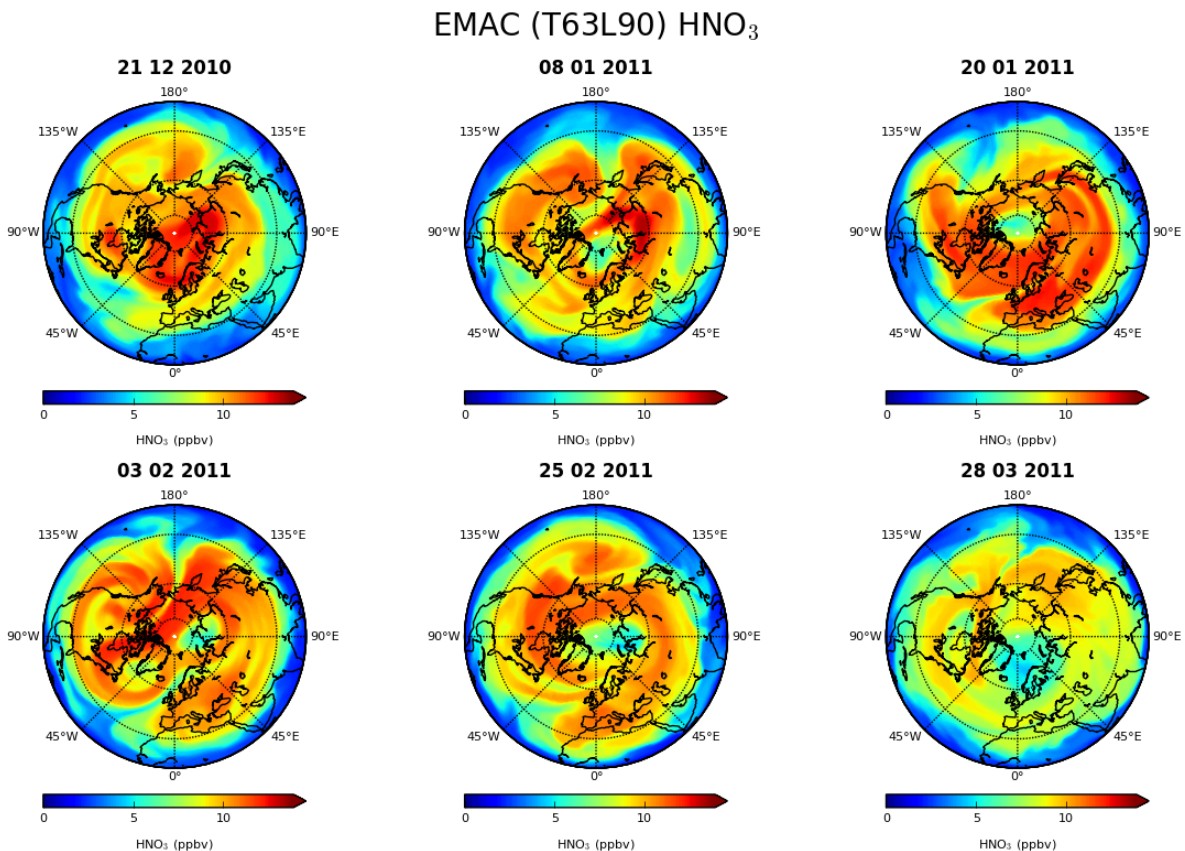

**Figure 7.** Distribution of $HNO_3$ as simulated with EMAC for 21 December 2010, 8 and 20 January 2011, 3 and 25 February and 28 March 2011 at $34\,hPa$ ($\approx 21\,km$).





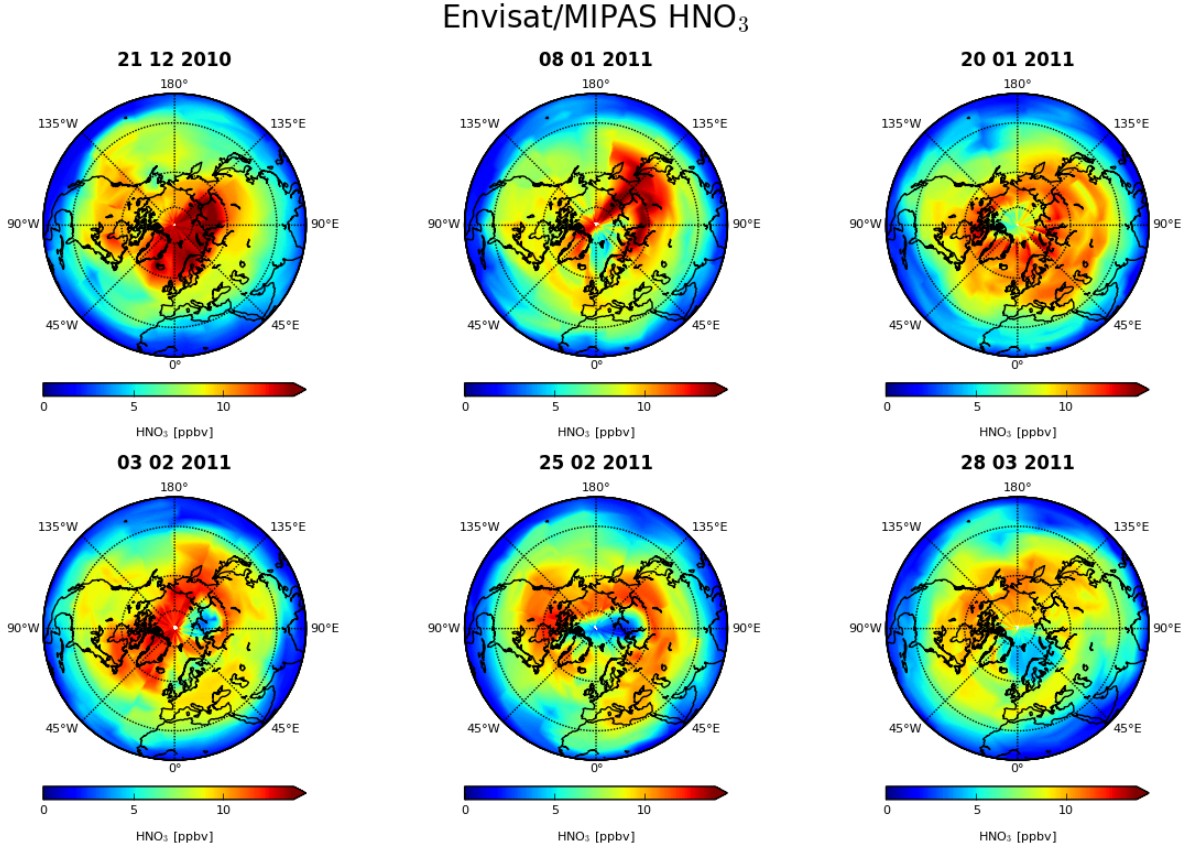

**Figure 8.** Distribution of HNO$_3$ as observed by Envisat/MIPAS for 21 December 2010, 8 and 20 January 2011, 3 and 25 February and 28 March 2011 at 21 km ($\sim$34 hPa).





**Figure 9.** Temporal evolution of temperature and $HNO_3$ at northern high latitudes (70-90°N) as function of pressure during the Arctic winter 2010/2011 (December 2010 to April 2011). Top: EMAC, middle: Envisat/MIPAS, bottom: Aura/MLS.





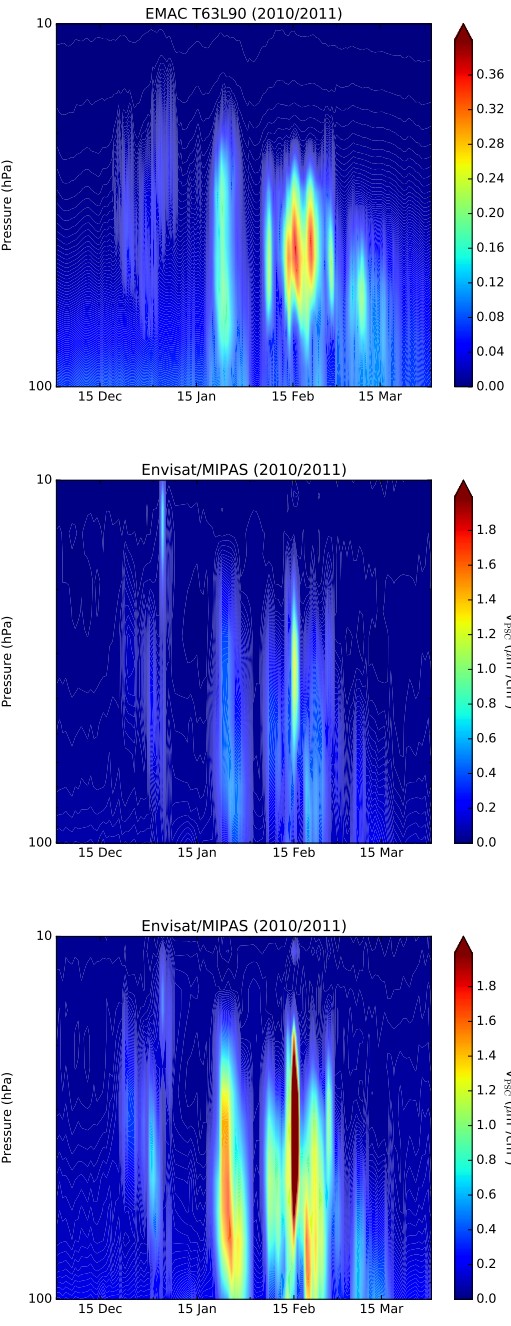

**Figure 10.** Volume density of all (liquid and solid) PSC particles ($V_{PSC}$) in $\mu m^3 cm^{-3}$ as simulated with EMAC (top) and observed with Envisat/MIPAS lower limit (middle) and Envisat/MIPAS upper limit (bottom) for the Arctic winter 2010/2011. Note the different colour bar scales between the Envisat/MIPAS and EMAC panels.



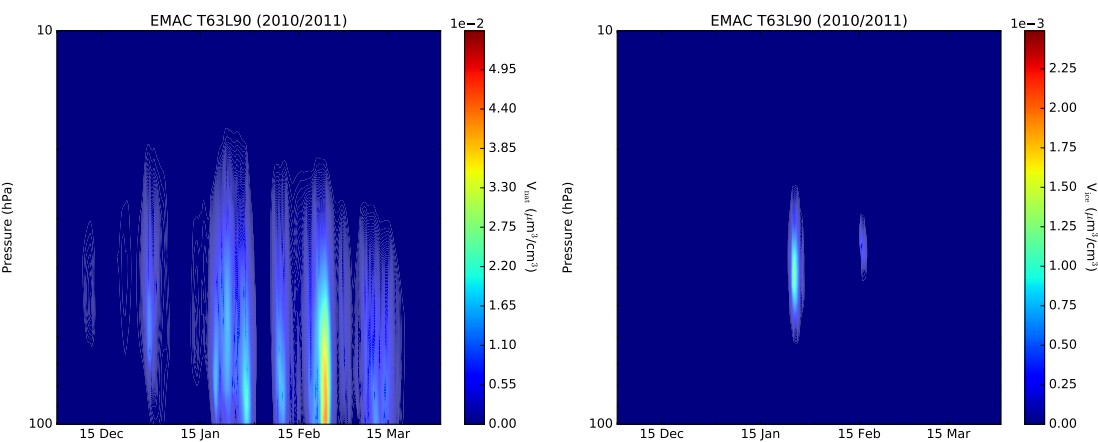

**Figure 11.** Volume density of solid particles ($V_{NAT}$ and $V_{ice}$) in $\mu m^3 cm^{-3}$ as simulated with EMAC for the Arctic winter 2010/2011.





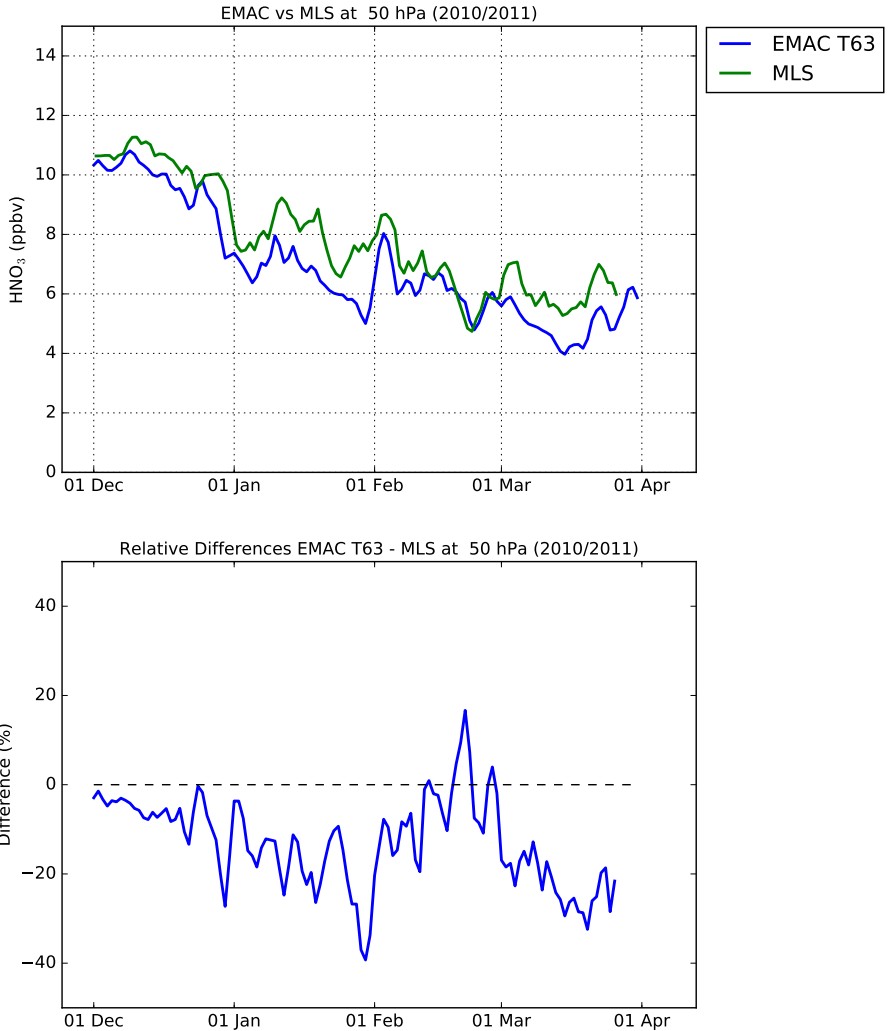

**Figure 12.** Top: Time series of $HNO_3$ (at 50 hPa) from EMAC (blue) and MLS (green) for the Arctic winter 2010/2011 (1 December 2011 to 30 April 2011, 70-90°N). Bottom: Relative differences of the EMAC - MLS time series.