# Peer review of "Comparison of ECHAM5/MESSy Atmospheric Chemistry (EMAC) Simulations of the Arctic winter 2009/2010 and 2010/2011 with Envisat/MIPAS and Aura/MLS Observations"

_Atmospheric Chemistry and Physics, 2017_

## Referee Comment (RC1) · Anonymous Referee #1 · 13 Feb 2018

General Comments

In this new study, Khosrawi et al. evaluate simulation results of the chemistry-climate model EMAC for the Arctic winters in 2009/2010 and 2010/2011. Simulation results for temperature, HNO3, and PSC volume density are compared with Envisat/MIPAS and Aura/MLS satellite observations.

Overall, the study fits in the scope of ACP and the manuscript is well written. However, I have a number of general and specific comments, which I recommend to be addressed

before the paper will be published.

1) In this paper you are not showing any evaluation results for stratospheric polar ozone. The fact that ozone data are available from the satellites for direct validation, but ozone simulation results are not discussed at all makes me wonder whether the simulated ozone distributions are far off reality (because the simulated PSC concentrations are too low by a factor of about 5 to 10)?

2) MIPAS also provides measurements of long-lived tracers such as N2O, CH4, and CFCs. Comparing simulation results for these tracers with the satellite observations may help to assess the proper representation of transport and mixing in EMAC.

3) The study finds good agreement between temperature distributions from EMAC and the satellite observations. However, the EMAC simulations have been nudged to ERA-Interim. From the paper I could not infer whether the nudging was rather weak or strong? Do the remaining temperature differences tell us something about the EMAC model or about the differences between the satellites and ERA-Interim?

Specific Comments

p7, l3-8: MIPAS PSC measurements during the Arctic winter 2010/11 are also discussed in a new ACPD paper by Spang et al. (2017).

p9, l14-16: The phrase saying "... PSC volume density is several orders of magnitude smaller..." is misleading. An order of magnitude refers to a change of a factor of 10. Several orders of magnitude may refer to change of a factor of 100, 1000, etc., but not to factors of 3 or 6-7 found here. This also needs to be fixed in other places.

p12, l30-32: Additionally, there might be problems because the standard flux-form semi-Lagrangian transport scheme in EMAC may be too diffusive near the transport barriers (Hoppe et al., 2014).

Figs. 6 and 12: Adding curves for Envisat/MIPAS may help to put the differences between MLS and the model into context.

Technical Corrections

p6, l2: "Arctic winter" -> "Arctic winters"

p6, l14: "orographic waves" -> "orographic gravity waves" (?)

p11, l25: "PSC seasons" -> "PSC phases" (?)

References

Hoppe, C. M., Hoffmann, L., Konopka, P., Grooß, J.-U., Ploeger, F., Günther, G., Jöckel, P., and Müller, R.: The implementation of the CLaMS Lagrangian transport core into the chemistry climate model EMAC 2.40.1: application on age of air and transport of long-lived trace species, Geosci. Model Dev., 7, 2639-2651, https://doi.org/10.5194/gmd-7-2639-2014, 2014.

Spang, R., Hoffmann, L., Müller, R., Grooß, J.-U., Tritscher, I., Höpfner, M., Pitts, M., Orr, A., and Riese, M.: A climatology of polar stratospheric cloud composition between 2002 and 2012 based on MIPAS/Envisat observations, Atmos. Chem. Phys. Discuss., https://doi.org/10.5194/acp-2017-898, in review, 2017.
* * *

---

## Referee Comment (RC2) · Anonymous Referee #2 · 26 Mar 2018

The authors present a detailed comparison of the Arctic atmosphere using the EMAC model system and the satellite instruments MIPAS on Envisat and MLS on EOS-Aura. The authors chose for the comparison two winters 2009/10 and 2010/11 on the grounds that both were extreme but showing different dynamic features. The publication is a description of the performance of the EMAC model in these two winters in the Arctic. The authors discuss differences in HNO3 in the comparison, but do only speculate about the causes, or cite other work which explains the deficiencies.

Therefore I wonder, if this publication is in the focus of ACP. From the 'Aims and Scope'

section, it seems to fit more a journal like 'Ceoscientific Model Development'.

The manuscript is generally well written. However, sometimes the wording is a bit sloppy. An example is the sentence: 'The Arctic winter 2010/2011 was one of the most persistently cold winters on record.' I guess, the authors want to emphasize that the stratosphere has been exceptionally cold, not the whole atmosphere or the troposphere alone.

The manuscript should be published, after the following minor issues have been addressed:

Please use a more stringent language. One example has been given above, but there are more throughout the manuscript.

Page 7 line 18-20

I cannot follow the argumentation why for the maps the 34hPa level have been used and for the time series comparison the 50 hPa level. Also, for the two levels, different satellite instruments have been used. Given this information I cannot help wondering, why no MLS maps and no MIPAS time series have been shown. The authors mentioned, that maps from MLS compare similar to EMAC, but I would suggest to put those figures in the supplement.

Figure 3 and 9 From figures 3 and 9, MLS and MIPAS show differences in the $HNO_3$ time series, but I would like to see, if it is mainly scaling or if there are different features. Although the authors put forward the argument that MIPAS is not sensitive to gasphase $HNO_3$ (page 8 line 15) in the presence of PSC's, the MIPAS measurements seem always higher than the MLS measurements.

The authors did not mention if the models have been convoluted with the AVK of the measurements. From the AURA/MLS quality document, the maximum AVK peak for $HNO_3$ is 0.8. Does this make a difference to the comparison in figures 3 and 9?

Figure 6 and 12.

[Figure]

The time series of HNO3 is discussed with frequent reference to the PSC occurrence in figures 4,5 and 10,11. However, given the different scaling and labeling of the x-axis of those figures, it is difficult to judge the authors conclusion about the differences in the model versus measurement comparison. I would suggest to align the scaling of the figures 4,5,6 and figures 10,11,12 and to put grid lines on the figures. This would make the reading much easier.

Page 8 line 18 In page 8, line 18 the authors write, that the concentration of HNO3 seems generally low by 2 ppbv throughout the winter, as can be seen in Figure 6 and 12. I find this difficult to see. In figure 6, EMAC and MLS do not differ anymore from mid-January till end of the time series. Later page 10, line 9 to 11, they restrict the difference to the PSC season. Please align the statements.

───────────────────────────

---

## Author Comment (AC1) · 2 May 2018

We thank reviewer 1 for the constructive, helpful criticism and the suggestion for revision. We followed the suggestions of reviewer 1 and revised the manuscript accordingly. Figures that we refer to in our answers to the referee comments are provided in the supplement to this reply.

*General Comments*
*In this new study, Khosrawi et al. evaluate simulation results of the chemistry-climate model*

[Figure]

*EMAC for the Arctic winters in 2009/2010 and 2010/2011. Simulation results for temperature, HNO₃, and PSC volume density are compared with Envisat/MIPAS and Aura/MLS satellite observations. Overall, the study fits in the scope of ACP and the manuscript is well written. However, I have a number of general and specific comments, which I recommend to be addressed before the paper will be published.*

*1) In this paper you are not showing any evaluation results for stratospheric polar ozone. The fact that ozone data are available from the satellites for direct validation, but ozone simulation results are not discussed at all makes me wonder whether the simulated ozone distributions are far off reality (because the simulated PSC concentrations are too low by a factor of about 5 to 10)?*

It is correct that the differences we find between the simulated PSCs and the ones observed by Envisat/MIPAS also affect the simulated $O_3$ distribution. Although we did not explicitly discuss other trace gases in the current paper, we have of course also evaluated some other gases. For example, for $O_3$ we find a generally good agreement between the EMAC simulations and Aura/MLS observations throughout the stratosphere (see Figure 1 and 3 in the supplement). Considering time series at 50 hPa (see Figure 2 and 4 in the supplement), we find at the beginning of the winter a very good agreement between EMAC and the Aura/MLS observations, but during the course of the winter when ozone destruction and descent become important, we find an increase of the differences, but these nevertheless do not exceed more than 20% (2009/2010) or 30% (2010/2011), respectively. We would like to keep the current study restricted to PSC volume density, temperature and HNO₃ since we presented already some (qualitative) comparisons of ozone simulated with EMAC to observations in Khosrawi et al. (2017). We referred to these comparisons presented in Khosrawi et al. (2017) in the introduction but repeat this now also in section 4 to remind the reader about this additional study.

*2) MIPAS also provides measurements of long-lived tracers such as N₂O, CH₄, and*

*CFCs. Comparing simulation results for these tracers with the satellite observations may help to assess the proper representation of transport and mixing in EMAC.*

We have also performed comparison between the $N_2O$ distribution simulated by EMAC and observations from Aura/MLS and found (in agreement with previous studies) that descent is underestimated (see Figure 5 and 6 in the supplement, stronger downward transport of low $N_2O$ mixing ratios is observed by Aura/MLS than is simulated with EMAC). To provide a thorough assessment of transport and mixing using several long-lived trace gases and adequate diagnostics for the assessment is quite complex and thus beyond the scope of this study. However, this will be the focus of future studies with EMAC.

*3) The study finds good agreement between temperature distributions from EMAC and the satellite observations. However, the EMAC simulations have been nudged to ERA-Interim. From the paper I could not infer whether the nudging was rather weak or strong? Do the remaining temperature differences tell us something about the EMAC model or about the differences between the satellites and ERA-Interim?*

In fact, the simulated temperatures are quite close to the ones derived from ERA-Interim since nudging is rather strong. Therefore, the differences found here tell us something about both the differences between the satellite observations and ERA-Interim as well as between the satellite observations and EMAC since both the model performance together with the data set used for nudging define the model results (see temperature comparisons between EMAC, ERA-Interim and satellite observations discussed in Jöckel et al. (2006) and Jöckel et al. (2016)). For example temperature fluctuations due to gravity waves are poorly resolved in the reanalysis data und thus the effect of gravity waves is underestimated in the model simulation but will be resolved by the satellite observations as e.g. MIPAS (Höpfner et al., 2006). When discussing the EMAC temperatures we added "nudged toward ECMWF ERA-Interim" to emphasize that these are nudged temperatures. Further, we added the following sentence in Sect. 4.1: *Note, due to the nudging the simulated temperatures are*

[Figure]

*quite close to those from ERA-Interim but some differences remain (see comparisons between EMAC, ERA-Interim and satellite observations presented in Jöckel et al. (2006) and Jöckel et al. (2016)). The influence of gravity waves on temperature, which is measured by the satellites but still not fully resolved in the reanalyses data (and thus not reflected in the nudged model simulation), is e.g. one reason why differences between model simulations and observations are found, not only with regard to temperature but also with regard to PSCs, see Höpfner et al., (2006).*

*Specific Comments*
*p7, l3-8: MIPAS PSC measurements during the Arctic winter 2010/11 are also discussed in a new ACPD paper by Spang et al. (2017).*
We added the following sentences referring to Spang et al. (2017) (now published in ACP) to this paragraph: *During the Arctic winter 2010/2011, the overall PSC occurrence frequency and area was exceptional for an Arctic winter as was shown by Manney et al. (2011) and confirmed in a recent study by Spang et al. (2018). The peak values of PSC area (in km$^2$) reached sizes comparable to June conditions in the Antarctic.*

*p9, l14-16: The phrase saying "... PSC volume density is several orders of magnitude smaller..." is misleading. An order of magnitude refers to a change of a factor of 10. Several orders of magnitude may refer to change of a factor of 100, 1000, etc., but not to factors of 3 or 6-7 found here. This also needs to be fixed in other places.*
Thanks for pointing this out. The relevant text phrases have been rewritten.

*p12, l30-32: Additionally, there might be problems because the standard flux-form semi-Lagrangian transport scheme in EMAC may be too diffusive near the transport barriers (Hoppe et al., 2014).*
This is correct, thanks for pointing this out. We added the following sentence referring to the study by Hoppe et al. (2014): *Furthermore, in a more recent study by Hoppe et al. (2014) it was shown that the standard flux-form semi Lagrangian scheme used in EMAC*
*may be too diffusive near the transport barriers and that the results can be improved when a Lagrangian transport scheme is used instead.*

*Figs. 6 and 12: Adding curves for Envisat/MIPAS may help to put the differences between MLS and the model into context.*

In fact, we had this in an earlier version of the paper and decided against it since it makes the discussion more complicated. Although the time series from Aura/MLS and Envisat/MIPAS are quite similar they are not the same. Using both satellites in the time series plots puts us in the need to discuss also the differences between the satellites which is however not the focus of this paper and distracts from what we actually want to discuss. However, due to a similar comment by reviewer 2 we added in the figure supplement to reviewer 2 a figure showing the time series from Aura/MLS vs Envisat/MIPAS for the Arctic winter 2010/2011. Further, in the manuscript we refer now to the study by Sheese et al. (2017) where a thorough comparison and discussion of the differences between MIPAS and MLS can be found.

*Technical Corrections*
*p6, l2: "Arctic winter" → "Arctic winters"*
*p6, l14: "orographic waves" → "orographic gravity waves" (?)*
*p11, l25: "PSC seasons" → "PSC phases" (?)*
Indeed, on p6, l14 it should read *orographic gravity waves* and on p11, l25 *PSC phases*. This has been corrected.

*References*
*Hoppe, C. M., Hoffmann, L., Konopka, P., Grooß, J.-U., Ploeger, F., Günther, G., Jöckel, P., and Müller, R.: The implementation of the CLaMS Lagrangian transport core into the chemistry climate model EMAC 2.40.1: application on age of air and transport of long-lived trace species, Geosci. Model Dev., 7, 2639-2651, https://doi.org/10.5194/gmd-7-2639-2014, 2014.*

[Figure]

*Spang, R., Hoffmann, L., Müller, R., Grooß, J.-U., Tritscher, I., Höpfner, M., Pitts, M., Orr, A., and Riese, M.: A climatology of polar stratospheric cloud composition between 2002 and 2012 based on MIPAS/Envisat observations, Atmos. Chem. Phys., https://doi.org/10.5194/acp-18-5089-2018, 2018.*

Please also note the supplement to this comment:
https://www.atmos-chem-phys-discuss.net/acp-2017-1190/acp-2017-1190-AC1-supplement.pdf

**Supplement:**

**Reply to Referee 1 Comments - Figures**

Manuscript-No: acp-2017-1190

**Comparison of ECHAM5/MESSy Atmospheric Chemistry (EMAC) Simulations of the Arctic winter 2009/2010 and 2010/2011 with Envisat/MIPAS and Aura/MLS Observations**

[Figure]

Figure 1: Temporal evolution of daily mean $O_3$ at northern high latitudes (averaged over 70-90°N) as function of pressure as simulated by EMAC T63L90 and observed by Aura/MLS for the Arctic winter 2009/2010.

[Figure]

Figure 2: Time series of O$_3$ from Aura/MLS measurements (green) and from the EMAC T63L90 (blue) simulation at ∼50 hPa averaged over 70-90°N for the Arctic winter 2009/2010.

[Figure]

Figure 3: Temporal evolution of daily mean $O_3$ at northern high latitudes (averaged over 70-90°N) as function of pressure as simulated by EMAC T63L90 and observed by Aura/MLS for the Arctic winter 2010/2011.

[Figure]

Figure 4: Time series of O$_3$ from Aura/MLS measurements (green) and from the EMAC T63L90 (blue) simulation at ∼50 hPa averaged over 70-90°N for the Arctic winter 2010/2011.

[Figure]

Figure 5: Temporal evolution of daily mean N$_2$O at northern high latitudes (averaged over 70-90°N) as function of pressure as simulated by EMAC T63L90 and observed by Aura/MLS for the Arctic winter 2009/2010.

[Figure]

Figure 6: Temporal evolution of daily mean $N_2O$ at northern high latitudes (averaged over 70-90°N) as function of pressure as simulated by EMAC T63L90 and observed by Aura/MLS for the Arctic winter 2010/2011.

---

## Author Comment (AC2) · 2 May 2018

We thank reviewer 2 for the constructive, helpful criticism and the suggestions for revision. We revised the manuscript according to the suggestions made. Additional to this reply, we provide a supplement with two figures, one showing the daily maps from Aura/MLS and the other one showing time series from Aura/MLS compared to Envisat/MIPAS for the Arctic winter 2010/2011.

*The authors present a detailed comparison of the Arctic atmosphere using the EMAC*

[Figure]

*model system and the satellite instruments MIPAS on Envisat and MLS on EOS-Aura. The authors chose for the comparison two winters 2009/10 and 2010/11 on the grounds that both were extreme but showing different dynamic features. The publication is a description of the performance of the EMAC model in these two winters in the Arctic. The authors discuss differences in $HNO_3$ in the comparison, but do only speculate about the causes, or cite other work which explains the deficiencies.*

The intention of this study is to evaluate the model performance concerning PSC formation and redistribution of $HNO_3$. Thus, this study is a quality assessment of the model performance and deficiencies in the model performance have been detected and discussed. Of course we try to explain these, but to solve the problems and improve the model performance within the frame of this study is beyond the scope and should thus be done in the frame of future studies.

*Therefore I wonder, if this publication is in the focus of ACP. From the 'Aims and Scope' section, it seems to fit more a journal like 'Geoscientific Model Development'.*

Before the submission of our manuscript we have checked the "Aims and Scope" of both journals and came to the conclusion that our study definitely fits better into the "Aims and Scope" of ACP since our study does not focus solely on the model comparison (evaluation), but also on the characteristics of the two Arctic winters considered.

*The manuscript is generally well written. However, sometimes the wording is a bit sloppy. An example is the sentence: 'The Arctic winter 2010/2011 was one of the most persistently cold winters on record.' I guess, the authors want to emphasize that the stratosphere has been exceptionally cold, not the whole atmosphere or the troposphere alone. The manuscript should be published, after the following minor issues have been addressed:*

It is correct that we wanted to emphasize that the stratosphere was exceptionally cold. We agree that the wording in this specific example is not precise on which part of the atmosphere is meant. We changed the sentence as follows:

*The Arctic winter 2010/2011 was one of the most persistently cold stratospheric winters on record, leading to the strongest depletion of ozone measured in the Arctic (Manney et al., 2011).*

*Please use a more stringent language. One example has been given above, but there are more throughout the manuscript.*

We have thoroughly checked the manuscript and made improvements in the text accordingly.

*Page 17 line 18-20*
*I cannot follow the argumentation why for the maps the 34 hPa level have been used and for the time series comparison the 50 hPa level. Also, for the two levels, different satellite instruments have been used. Given this information I cannot help wondering, why no MLS maps and no MIPAS time series have been shown. The authors mentioned, that maps from MLS compare similar to EMAC, but I would suggest to put those figures in the supplement.*

We agree that the sentence is not precisely formulated. Gas phase depletion is pronounced over a certain pressure range and we just meant that in the chosen presentation forms (daily polar stratospheric maps and time series based on a 70-90°N latitude average) the gas phase depletion was most pronounced visible in the respective levels. Therefore, we changed the sentence as follows: *Comparisons of temperature, $HNO_3$, and PSC volume densities are shown over the full vertical range (200-6 hPa) in time-pressure cross sections. In addition, for $HNO_3$ we have chosen to highlight the 34 hPa level, where daily maps show the most pronounced gas-phase depletion features, and 50 hPa, where such signatures are largest in the time series of polar-cap averaged $HNO_3$ mixing ratios.*

In an earlier version of the paper we had the time series of both satellite instruments included in the figure, but decided in the end against it since it made the discussion of the results more complicated. Although the time series from Aura/MLS and Envisat/MIPAS are quite similar they are not the same. Using both satellites in the time series plots puts us in the need to discuss also the differences between the satellite

instruments which is, however, not the focus of this paper and distracts from what we actually want to discuss. The same holds for the daily maps. Showing both makes the discussion more complicated. By using MIPAS for the daily maps, both satellites for the contour plots and MLS in the time series resulted in a fair employment of the respective satellite data sets used in this study. We added the daily maps for Aura/MLS and a comparison of MLS and MIPAS time series for the Arctic winter 2010/2011 as a supplement to this reply (Figures 1 and 2), but think that for the final version of the paper such a supplement is not necessary.

*Figure 3 and 9 From figures 3 and 9, MLS and MIPAS show differences in the $HNO_3$ time series, but I would like to see, if it is mainly scaling or if there are different features. Although the authors put forward the argument that MIPAS is not sensitive to gasphase $HNO_3$ (page 8 line 15) in the presence of PSC's, the MIPAS measurements seem always higher than the MLS measurements.*

It is correct that the MIPAS $HNO_3$ mixing ratios are generally somewhat higher than the ones from Aura/MLS. We see that e.g. in the time series when we plot both satellite data sets (see Figure 2 in the supplement). Further, the difference in the $HNO_3$ mixing ratios between MLS and MIPAS is not solely related to the sampling bias of MIPAS in the presence of PSCs. Differences between these instruments are found during the entire course of the winter, but the differences increase when PSCs are present. The differences in $HNO_3$ between MIPAS and MLS have been documented in the literature. The paper by Sheese et al. (2017) discusses comparisons of ACE-FTS, MLS, and MIPAS profiles of several species. In all their figures that showed relative differences of $HNO_3$ profiles they found that MIPAS had a more positive bias than MLS in the altitude range 18–27 km, indicating that MIPAS provides higher $HNO_3$ values in general (their Figs. 7, second row, third panel from the right; 9, bottom right panel; 11, bottom right panel). We changed the the sentence at P8, L14 as follows: *The patterns of the temporal evolution of $HNO_3$ derived from Envisat/MIPAS and Aura/MLS are generally similar, however, Envisat/MIPAS provides somewhat higher $HNO_3$ abundances*

*than Aura/MLS (see e.g. Sheese et al. 2017 for a throrough comparison and discussion of the differences between Aura/MLS and Envisat/MIPAS).*

*The authors did not mention if the models have been convoluted with the AVK of the measurements. From the AURA/MLS quality document, the maximum AVK peak for HNO$_3$ is 0.8. Does this make a difference to the comparison in figures 3 and 9?*

We have not convolved the model data with the AVKs of the measurements. Although there are differences in the vertical resolution (EMAC: 1 km, MLS: 3-4 km, MIPAS: 3 km) we would nevertheless expect that differences between the model data and the model data convolved with the averaging kernels are small since we use averaged data (daily zonal means) which results already in some smoothing of the data. Further, we would rather expect that the usage of the averaging kernels from e.g. MLS would result in even lower HNO$_3$ mixing ratios and thus would increase the differences.

*Figure 6 and 12.*

*The time series of HNO$_3$ is discussed with frequent reference to the PSC occurrence in figures 4,5 and 10,11. However, given the different scaling and labeling of the x-axis of those figures, it is difficult to judge the authors conclusion about the differences in the model versus measurement comparison. I would suggest to align the scaling of the figures 4,5,6 and figures 10,11,12 and to put grid lines on the figures. This would make the reading much easier.*

On Figures 6 and 12 we have already grid lines in the upper panels and included now also grid lines to the lower panels. Putting grid lines on Figure 4, 5 and 10, 11, however, will not work. If we put grid lines on these plots these will not be clearly visible if these are in black. If we use white grid lines instead these will be too strong and make seeing the volume density distribution difficult. Adjusting the x-scaling for figures 6 and 12, however, is no problem. We have done this.

*Page 8 line 18 In page 8, line 18 the authors write, that the concentration of HNO$_3$ seems generally low by 2 ppbv throughout the winter, as can be seen in Figure 6 and 12. I find*

*this difficult to see. In figure 6, EMAC and MLS do not differ anymore from mid-January till end of the time series. Later page 10, line 9 to 11, they restrict the difference to the PSC season. Please align the statements.*

Thanks a lot for pointing this inconsistency out. We changed the sentence at page 8, line 18 as follows: *However, this comparison shows that EMAC HNO$_3$ seems generally 1-2 ppbv too low in early winter and throughout the PSC season.*

References:
Sheese, P. E., Walker, K. A., Boone, C. D., Bernath, P. F., Froidevaux, L., Funke, B., Raspollini, P., and von Clarmann, T., ACE-FTS ozone, water vapour, nitrous oxide, nitric acid, and carbon monoxide profile comparisons with MIPAS and MLS, J. Quant. Spectrosc. Radiat. Transfer, 186, 63-80, 2017.

Please also note the supplement to this comment:
https://www.atmos-chem-phys-discuss.net/acp-2017-1190/acp-2017-1190-AC2-supplement.pdf
* * *
[Figure]

**Supplement:**

**Reply to Referee 2 Comments - Figures**

Manuscript-No: acp-2017-1190

**Comparison of ECHAM5/MESSy Atmospheric Chemistry (EMAC) Simulations of the Arctic winter 2009/2010 and 2010/2011 with Envisat/MIPAS and Aura/MLS Observations**

[Figure]

Figure 1: Distribution of HNO$_3$ as observed by Aura/MLS for 21 December 2010, 3 and 25 February 2011 at 34 hPa ($\sim$21km). Note: no data for 28 March 2011.

[Figure]

Figure 2: Time series of HNO$_3$ (at 50 hPa) from MIPAS (blue) and Aura/MLS (green) for the Arctic winter 2010/2011 (1 December to 31 March, 70-90°N).

---

## Editor Decision (ED1)

**General**

1. The extreme winters are taken for testing the model performance, which is good and challenging (e.g. which is why the differences are up to 40—45%). However, the average/normal state should also be known as the extreme cases are "rare". Therefore, a discussion based on a normal (not very cold/warm) winter would have been a good addition to these analyses. If you have already a simulation, it would be helpful to include or at least mention the performance of the model for a normal year. Although you have mentioned about your previous work on Antarctic, please do mention the model's performance for the Antarctic region in the discussion section to make the sensitivity tests "complete".

2. You were also talking about the relationship between ozone loss and denitrification. However, nothing is done for this, as you were focussing completely on the latter. Therefore, please indicate how the ozone loss or chlorine activation is simulated in the model.

3. How was performance of this model in CCM val exercise? Please briefly mention that too, either in the introduction or in the discussion. Also, mention whether your sensitivity tests will improve the performance of the model for further assessments and reports. It is also good to include a general statement on this in the abstract.

**Technical corrections**

**Page 1:**
Line 2: delete ERA
Line 3: winters
Line 4: as previous studies
Line 5: for the Arctic winters
Line 9: largest deletion not strongest (strongest can also be episodes)
Line 13: occurred in that winter
Line 14: PSC formation and denitrification
Line 17: and associated sequestration
Line 18: smaller than that derived from
Line 18: Furthermore
Line 19: "as high as" , is this altitude? Or value?
Line 20: You need a concluding statement here on your model simulations, EMAC or the CCMs in general to put the results in perspective.
Line 21: winters

**Page 2**
Line 3, 7: very small sentences make the reading difficult
Line 17: largest depletion

Line 24-25: Sentence is not complete
Line 25: two e.g. are there in the same sentence. Please construct a better sentence.
Line 29: "and results ..."

**Page 4**
Line 11: delete further
Line 11: temperatures were

**Page 6**
Line 6: "Additionally there is no …". There is no connection. Why gravity waves are mentioned here.
Line 24: "ice PSCs and (4)"

**Page 7**
Line 10: data are "
Line 10: 13 march 2011

**Page 8**
Line 14/20: replace somewhat by slightly or give the value here
Line 25: mid-January?
Line 27: mid-January and mid-December?
Line 30—33: the "e.g" and "see" makes the sentence difficult to read. Please rephrase this.

**Page 10**
Line 21-22: Any idea how much that would make for chlorine activation and ozone loss estimation or simulation in the model?

**Page 13**
Line 6-7: This problem is common to most models, as I know. This is also reported in some other studies.

**Page 16**
Line 11—12: Please remove, if this is not published yet.

---

## Author Response (AR2)

**Reply to Editor Comments**

**Manuscript-No: acp-2017-1190**

**Comparison of ECHAM5/MESSy Atmospheric Chemistry (EMAC) Simulations of the Arctic winter 2009/2010 and 2010/2011 with Envisat/MIPAS and Aura/MLS Observations**

We thank the editor for the suggestion for revision. Please find our point by point reply below.

*General Comments*
*1. The extreme winters are taken for testing the model performance, which is good and challenging (e.g. which is why the differences are up to 40-45%). However, the average/normal state should also be known as the extreme cases are rare. Therefore, a discussion based on a normal (not very cold/warm) winter would have been a good addition to these analyses. If you have already a simulation, it would be helpful to include or at least mention the performance of the model for a normal year. Although you have mentioned about your previous work on Antarctic, please do mention the models performance for the Antarctic region in the discussion section to make the sensitivity tests complete.*

We have also performed comparisons for the rather warm Arctic winter 2008/2009. For the comparison of $HNO_3$ a better agreement is found throughout the winter simply due to the fact that the PSC season is shorter. The development throughout the course of the winter of e.g. $HNO_3$ time series is the same as for the cold winters, differences in $HNO_3$ between model simulation and observation increase as soon as PSC are present, but differences do not get as large as for the cold winters due to the smaller amount of PSCs. We added the following sentence in the conclusion: *Since we have chosen two extreme winter for this study, the here derived differences are the largest possible differences since not all Arctic winters are extreme. Nevertheless, the here discussed issues in the model performance remain, but differences between model simulation and observations are throughout the PSC season not as large for a rather warm winter, as e.g the Arctic winter 2008/2009.*

We added the following sentence in the conclusion discussing the earlier study on the Antarctic by Kirner et al. (2015): *In the study by Kirner et al. (2015) only qualitative comparisons of ClO, $O_3$ and $HNO_3$ to Aura/MLS measurements were performed mainly based on multi-year averages.* However, it is not useful to discuss/mention their results since they had a completely different approach than we have (using multi-year averages while we focus on the performance of a single winter). Further, their simulations are based on an older version of the model while we use a newer version that has been significantly improved.

*2. You were also talking about the relationship between ozone loss and denitrification. However, nothing is done for this, as you were focussing completely on the latter. Therefore, please indicate how the ozone loss or chlorine activation is simulated in the model.*

We added now the following sentences in the conclusion: *As found in Khosrawi et al. (2017) for the Arctic winter 2015/2016, also here a very good agreement between the simulated and measured $O_3$ is found at the beginning of the winter. However, during the course of the winter when ozone destruction and descent become important, an increase of the differences is found. These however do not exceed 20% (2009/2010) or 30% (2010/2011), respectively.*

*3. How was performance of this model in CCM val exercise? Please briefly mention that too, either in the introduction or in the discussion. Also, mention whether your sensitivity tests will improve the performance of the model for further assessments and reports. It is also good to include a general statement on this in the abstract.*

Also here, it is unfortunately not very useful to discuss the performance of EMAC in the frame of CCM val since also here the simulations were performed with an older version of the model and the comparisons were focusing on climatological aspects rather than the performance of a single winter. Nevertheless, the here derived results will help to improve the setup of the model and thus the simulations which will be of benefit for future model intercomparison studies. Therefore, we changed the last sentence of the conclusions as follows: *Further, the here derived results and upcoming sensitivity simulations will serve as benchmark for the development of the PSC parameterisation in other atmospheric models as e. g. ICON-ART (ICOsahedral Nonhydrostatic Model - Aerosols and Reactive Trace gases) and will help to improve the performance of EMAC in future model intercomparison studies.* In the abstract we added the following sentence: *The here found differences between model simulations and observations stipulate farther improvements in the EMAC set-up for simulating PSCs.*

*Technical corrections*
*Page 1:*
*Line 2: delete ERA*
Why? we have used ERA-Interim reanalyses, so therefore this should also be stated in the manuscript.

*Line 3: winters*
This has been corrected.

*Line 4: as previous studies*
This has been changed as suggested.

*Line 5: for the Arctic winters*
In line 5 it should read Antarctic and not Arctic. We have changed the sentence as follows: *This study is the first to perform an extensive assessment of the performance of the EMAC model for Arctic winters as previous studies have only made limited evaluations of EMAC simulations which also were mainly focused on the Anatarctic winter stratosphere..*

*Line 9: largest depletion not strongest (strongest can also be episodes)*
We would prefer to keep *"strongest"*. We used also *"strongest"* in our 2017 ACP paper (that went through copy-editing and is published).

*Line 13: occurred in that winter*
This has been changed as suggested.

*Line 14: PSC formation and denitrification*
This has been changed as suggested.

*Line 17: and associated sequestration*
This has been corrected as suggested.

*Line 18: smaller than that derived from*
We think *"smaller than the ones...."* is correct and would thus keep this text part as is.

*Line 18: Furthermore*
Here, it should not matter if one uses *"further"* or *"furthermore"*. We would prefer to keep *"further"* to be consistent with our way of writing.

*Line 19: as high as , is this altitude? Or value?*
Yes, as high as in altitude. We added now *in altitude* in parentheses to be more precise.

*Line 20: You need a concluding statement here on your model simulations, EMAC or the CCMs in general to put the results in perspective.*
We added the following sentence to the abstract: *The here found differences between model simulations and observations stipulate farther improvements in the EMAC set-up for simulating PSCs.*

*Line 21: winters*
In line 21 there is neither the word "winters" missing nor "s" in "winter". Therefore, we could not change anything with respect to this comment.

*Page 2*

*Line 3, 7: very small sentences make the reading difficult*
We have combined the small sentences to make the reading easier.

*Line 17: largest depletion*
As stated above we would like to keep *strongest*.

*Line 24-25: Sentence is not complete*
*Line 25: two e.g. are there in the same sentence. Please construct a better sentence.*
We changed the text as follows and hope that these two sentences are clearer now: *The Arctic winter 2009/2010 has been well analysed both by measurements and model simulations. For example, detailed studies on denitrification during this winter were performed by e.g. Khosrawi et al. (2011) and on dehydration by e.g. Khaykin et al. (2013).*

*Line 29: and results ...*
This has been changed as suggested.

*Page 4*
*Line 11: delete further*
*Line 11: temperatures were*
We could not find the respective text parts at the page and line number given. Therefore, nothing has been changed here. Anyway, this should be no problem since the copy-editing will correct all remaining language errors.

*Page 6*
*Line 6: Additionally there is no .... There is no connection. Why gravity waves are mentioned here.*
During this winter PSCs were formed by both, synoptic-scale cooling and by gravity waves. In the Arctic usually gravity waves are required to initiate the formation of ice PSCs (otherwise temperatures do not get as cold as required for ice formation). However, this winter was quite unusual, because the synoptic cooling was quite strong during this winter and was responsible for almost all ice PSCs. Gravity waves occurred as well, but played in this winter only a minor role.

*Line 24: ice PSCs and (4)*
We are not sure if the suggested correction is correct or required. Therefore, we would leave that as it is and see what the copy-editing does in this case.

*Page 7*
*Line 10: data are*
*Line 10: 13 march 2011*
Line 10 has been corrected as suggested.

*Page 8*
*Line 14/20: replace somewhat by slightly or give the value here*
We have replaced *"somewhat"* by *"slightly"*.

*Line 25: mid-January?*
*Line 27: mid-January and mid-December?*
Yes, thanks! This has been corrected.

*Line 3033: the e.g and see makes the sentence difficult to read. Please rephrase this.*
We agree and we give now the reference of Höpfner et al. (2006) in parentheses.

*Page 10*
*Line 21-22: Any idea how much that would make for chlorine activation and ozone loss estimation or simulation in the model?*
The underestimation of denitrification also influences the simulation of chlorine activation and ozone loss, but is difficult to quantify since also other factors play a role like the underestimation of transport in the model and inaccuracies in the partitioning between chlorine species at high solar zenith angles as was discussed in Khosrawi et al. (2017). We added the following sentence at the end of section 4.1: *The underestimation of $HNO_3$ in the model simulation affects also chlorine activation and ozone loss, but also other factors like the underestimation of transport in the model and inaccuracies in the partitioning between chlorine species at high solar zenith angles as discussed for the Arctic winter 2015/2016 in Khosrawi et al. (2017) play a role.*
*Page 13*
*Line 6-7: This problem is common to most models, as I know. This is also reported in some other studies.*
We do not know of any study really discussing this issue. All studies we know of have their focus on the simulation of the occurrence frequency of the SSWs, but not if the strength of the SSW is correctly simulated.

*Page 16*
*Line 1112: Please remove, if this is not published yet.*
The paper by Höpfner et al. (2018) has been submitted to AMT and should be published soon.